# SIMILARITY OF NEURAL ARCHITECTURES BASED ON INPUT GRADIENT TRANSFERABILITY

## ABSTRACT

In this paper, we aim to design a quantitative similarity function between two neural architectures. Specifically, we define a model similarity using input gradient transferability. We generate adversarial samples of two networks and measure the average accuracy of the networks on adversarial samples of each other. If two networks are highly correlated, then the attack transferability will be high, resulting in high similarity. Using the similarity score, we investigate two topics: (1) Which network component contributes to the model diversity? (2) How does model diversity affect practical scenarios? We answer the first question by providing feature importance analysis and clustering analysis. The second question is validated by two different scenarios: model ensemble and knowledge distillation. We conduct a large-scale analysis on 69 state-of-the-art ImageNet classifiers. Our findings show that model diversity takes a key role when interacting with different neural architectures. For example, we found that more diversity leads to better ensemble performance. We also observe that the relationship between teacher and student networks and distillation performance depends on the choice of the base architecture of the teacher and student networks. We expect our analysis tool helps a high-level understanding of differences between various neural architectures as well as practical guidance when using multiple architectures.

## 1 INTRODUCTION

The last couple of decades have seen the great success of deep neural networks (DNNs) in real-world applications, *e.g.*, image classification (He et al., 2016a) and natural language processing (Vaswani et al., 2017). The advances in the DNN architecture design have taken a key role in this success by making the learning process easier (*e.g.*, normalization methods (Ioffe & Szegedy, 2015; Wu & He, 2018; Ba et al., 2016) or skip connection (He et al., 2016a)) enforcing human inductive bias in the architecture (*e.g.*, convolutional neural networks (CNNs) (Krizhevsky et al., 2012; Simonyan & Zisserman, 2015)) or increasing model capability with the self-attention mechanism (*e.g.*, Transformers (Vaswani et al., 2017)). With different design principles and architectural elements, there have been proposed a number of different neural architectures; however, designing a distinguishable architecture is expensive and needs heavy expertise. One of the reasons for the difficulty is that there is only a little knowledge of the difference between two different neural architectures. Especially, if one can quantify the similarity between two models, then we can measure which design component actually contributes significantly to diverse properties of neural networks. We also can utilize the quantity for new model design (*e.g.*, by neural architecture search (NAS) (Zoph & Le, 2017)).

In this paper, we aim to define the similarity between two networks to quantify the difference and diversity between neural architectures. Existing studies have focused on dissecting each network component layer-by-layer (Kornblith et al., 2019; Raghu et al., 2021) or providing a high-level understanding by visualization of loss surface (Dinh et al., 2017), input gradient (Springenberg et al., 2015; Smilkov et al., 2017), or decision boundary (Somepalli et al., 2022). On the other hand, we aim to design an architecture agnostic and quantitative score to measure the difference between the two architectures. We especially focus on the input gradients, a widely-used framework to understand model behavior, e.g., how a model will change predictions by local pixel changes (Sung, 1998; Simonyan et al., 2014; Springenberg et al., 2015; Smilkov et al., 2017; Sundararajan et al., 2017; Bansal et al., 2020; Choe et al., 2022). If two models are similar, then the input gradients are similar. However, because an input gradient is very noisy, directly measuring the difference between input

gradients is also very noisy. Instead, we use the *adversarial attack transferability* as the proxy measure of the difference between input gradients of two networks. Consider two models $A$ and $B$, and the input $x$. We generate adversarial samples $x_A$ and $x_B$ to $A$ and $B$, respectively. Then we measure the accuracy of $B$ for $x_A$ ($\text{acc}_{A \to B}$) and vice versa. If $A$ and $B$ are similar and assume an optimal adversary, then $\text{acc}_{A \to B}$ will be almost zero, while if $A$ and $B$ have distinct input gradients, then $\text{acc}_{A \to B}$ will not be dropped significantly. We also note that adversarial attack transferability will provide a high-level understanding of the difference between model decision boundaries (Karimi & Tang, 2020). We define a model similarity based on attack transferability and analyze the existing neural architectures, *e.g.*, which network component affects the model diversity the most?

We measure the pairwise attack transferability-based network similarity of 69 different neural architectures trained on ImageNet (Russakovsky et al., 2015), provided by Wightman (2019). Our work is the first extensive study for model similarity on a large number of state-of-the-art ImageNet models. Our first goal is to understand the effect of each network module on model diversity. We first choose 13 basic components (*e.g.*, normalization, activation, the design choice for stem layers) that consist of neural architectures and list the components of 69 networks. For example, we represent ResNet (He et al., 2016a) as $f_{\text{ResNet}} = [\text{Base architecture} = \text{CNN}, \text{Norm} = \text{BN}, \text{Activation} = \text{ReLU}, \dots]$. We analyze the contribution of each network component to the model diversity by using the feature importance analysis by gradient boosting regression on model similarities, and the clustering analysis on model similarities. Our analyses show that the choice of base architecture (*e.g.*, CNN, Transformer) contributes most to the network diversity. Interestingly, our analysis shows that the design choice for the input-level layers (*e.g.*, stem layer design choice) determines the network diversity as much as the choice of core modules (*e.g.*, normalization layers, activation functions).

Our study is not only limited to the understanding of component-level architecture design, but we also focus on analyzing the effect of model diversity in practical scenarios. Particularly, we measure the *model ensemble performances* by controlling the diversity of the candidate models, *e.g.*, ensemble "similar" or "dissimilar" models by our similarity score. Here, we observe that when we ensemble more dissimilar models, the ensemble accuracy gets better; *more diversity leads to a better ensemble performance*. We observe that the diversity caused by different initialization, different hyper-parameter choice, and different training regimes is not as significant as the diversity caused by architecture change. We also observe that the ensemble of models from the same cluster performs worse than the ensemble of models from different clusters. Similarly, by choosing more diverse models, the number of wrong samples by all models is getting decreased. Our findings confirm that the previous study conducted in simple linear classifiers (Kuncheva & Whitaker, 2003) is also aligned with recent complex large-scale neural networks.

As our third contribution, we provide a practical guideline for the choice of a teacher network for knowledge distillation (KD) (Hinton et al., 2015). We train 25 distilled `ViT-Ti` models with diverse teacher networks. Interestingly, our findings show that the performance of the distilled model is highly correlated to the similarity between the teacher and the student networks rather than the accuracy of the teacher network; if the student networks and the teacher network are based on the same architecture (*e.g.*, both are based on Transformer), then a similar teacher provides better knowledge; if the student and teacher networks are based on different architectures (*e.g.*, Transformer and CNN), then we observe that choosing a more dissimilar teacher will lead to a better distillation performance. Our findings are partially aligned with previous KD studies, *i.e.*, using a similar teacher network leads to a better KD performance (Jin et al., 2019; Mirzadeh et al., 2020). However, the existing studies only focus on the scenario when both teacher and student networks are based on the same architecture, exactly aligned with our findings. Our observation extends previous knowledge to the case of when the teacher and student networks significantly differ (*e.g.*, Transformer and CNN).

## 2  NETWORK SIMILARITY BY INPUT GRADIENT TRANSFERABILITY

In this section, we propose a similarity measure between two networks based on adversarial attack transferability. Our interest lies in developing a practical toolbox to measure the architectural difference between the two models quantitatively. Existing studies for a similarity between deep neural networks have focused on comparing intermediate features (Kornblith et al., 2019; Raghu et al., 2021), understanding loss landscapes (Dinh et al., 2017; Li et al., 2018; Park & Kim, 2022), or decision boundary (Somepalli et al., 2022). However, their approaches cannot measure the distance

between two different networks (Li et al., 2018; Kornblith et al., 2019) or need expensive computation costs for re-training networks multiple times (Somepalli et al., 2022). There also has been another line of research based on prediction-based statistics, *e.g.*, comparing the wrong and correct predictions (Kuncheva & Whitaker, 2003; Geirhos et al., 2018; 2020; Scimeca et al., 2022). Although this approach can be used for measuring a model diversity (Kuncheva & Whitaker, 2003), they only can capture the prediction itself, not the change of prediction. However, just focusing on prediction values could be misleading. Because recent deep models are highly complex, the change of predictions (or sensitivity) is more practical than focusing on the prediction itself. For example, Geirhos et al. (2020) showed that recent complex deep neural networks show highly similar predictions. Also, a deep model easily changes its prediction with a neglected small noise (*e.g.*, adversarial attack), and the perturbation varies by different models, as shown in our experiments.

In this paper, we focus on how two models change their predictions by input changes, *e.g.*, the input gradient. Input gradient is a popular visualization tool for understanding DNNs by highlighting where a model attends (Sung, 1998; Simonyan et al., 2014). An input gradient-based method has several advantages; it is computationally efficient, and no additional training is required; it can provide a visual understanding of the given input. Usually, an input gradient-based quantity measures how the input gradient of a model matches the actual foreground (Choe et al., 2022), *i.e.*, it usually measures how a model explains the input well. Because we are interested in comparing two different models, we focus on how input gradients are similar between them. Especially, we generate an adversarial sample using a model and test the adversarial sample to the other architecture.

Adversarial attack transferability is also a good approximation of measuring the difference between model decision boundaries (Karimi & Tang, 2020). Comparing the decision boundaries of two models will provide a high-level understanding of how two models behave differently for input changes. Unfortunately, an exact decision boundary is not achievable, and an expensive and inexact sampling approximation is required (Somepalli et al., 2022). In this paper, we analyze the difference between two decision boundaries by generating an adversarial sample for each decision boundary. If two models have similar decision boundaries, then the adversarial samples will be transferred (*i.e.*, will have high adversarial attack transferability); if two models have dissimilar decision boundaries, then the adversarial samples will not be transferred. The detailed discussion can be found in Appendix H.

Formally, we generate adversarial samples $x_A$ and $x_B$ of model $A$ and $B$ for the given input $x$. Then, we measure the accuracy of model $A$ using the adversarial sample for model $B$ (called $\text{acc}_{B \to A}$). If $A$ and $B$ are the same, then $\text{acc}_{B \to A}$ will be zero (for simplicity, here, we may assume that the adversary can fool a model perfectly). On the other hand, if the input gradients of $A$ and $B$ differ significantly, then the performance drop will be neglectable because the adversarial sample is made almost similar to the original image (*i.e.*, $\|x - x_B\| \leq \varepsilon$). Let $X_{AB}$ be the set of inputs where both $A$ and $B$ predict correctly, $y$ be the ground truth label, and $\mathbb{I}(\cdot)$ be the indicator function. We use the following quantity to measure the similarity between two models:

$$s(A, B) = \log \left[ \max \left\{ \varepsilon_s, 100 \times \frac{1}{2|X_{AB}|} \sum_{x \in X_{AB}} \{\mathbb{I}(A(x_B) \neq y) + \mathbb{I}(B(x_A) \neq y)\} \right\} \right], \quad (1)$$

where $\varepsilon_s$ is a small scalar value to prevent the ill-posedness of $s(\cdot)$. If $A = B$ and we have an oracle adversary, then $s(A, A) = \log 100$. In practice, a strong adversary (*e.g.*, PGD (Madry et al., 2018) or AutoAttack (Croce & Hein, 2020)) can easily achieve a nearly-zero accuracy if a model is not trained by an adversarial attack-aware strategy, *e.g.*, adversarial training (Madry et al., 2018; Cohen et al., 2019). On the other hand, if the adversarial attacks on $A$ are not transferable to $B$ and vice versa, then $s(A, B) = \log \varepsilon_s$. We add more discussions how equation 1 can serve as a similarity score function from a functional point of view in Appendix A.

Now, we provide an extensive analysis of a large number of state-of-the-art networks using our quantity. Our analysis focuses on two questions. (1) Which network component contributes to the diversity between models? (2) How the model diversity affects the practical applications, *e.g.*, model ensemble or knowledge distillation? Note that several studies attempted to answer the second question in different ways (Kuncheva & Whitaker, 2003; Teney et al., 2022), but our work is the first work to analyze 69 state-of-the-art ImageNet models based on model similarity. As observed by previous works in a limited number of models without considering the change of architecture and large-scale training datasets, our findings show that diversity can contribute to practical applications in the state-of-the-art complex deep neural networks trained on large-scale datasets.

Table 1: **Overview of model elements.** We categorize each architecture with 13 different architectural components. The full feature list of each architecture is in Appendix B.

| Components | Elements |
|---|---|
| Base architecture | CNN, Transformer, MLP-Mixer, Hybrid (CNN + Transformer), NAS-Net |
| Stem layer | 7×7 conv with stride 2, 3×3 conv with stride 2, 16×16 conv with stride 16, ... |
| Input resolution | 224×224, 256×256, 240×240, 299×299 |
| Normalization layer | BN, GN, LN, LN + GN, LN + BN, Normalization-free, ... |
| Using hierarchical structure | Yes (*e.g.*, CNNs, Swin (Liu et al., 2021b)), No (*e.g.*, ViT (Dosovitskiy et al., 2021)) |
| Activation functions | ReLU, HardSwish, SiLU, GeLU, ReLU + GeLU, ReLU + SiLU or GeLU ... |
| Using pooling at stem | Yes, No |
| Using 2D self-attention | Yes (*e.g.*, Bello (2021), Srinivas et al. (2021), Vaswani et al. (2021)), No |
| Using channel-wise (CW) attention | Yes (*e.g.*, Hu et al. (2018), Cao et al. (2019), Wang et al. (2020c)), No |
| Using depth-wise convolution | Yes, No |
| Using group convolution | Yes, No |
| Type of pooling for final feature | Classification (CLS) token, Global Average Pool (GAP) |
| Location of CW attentions | At the end of each block, in the middle of each block, ... |

**Settings.** We use the projected gradient descent (PGD) for the adversary. We set the iteration to 50, the learning rate to 0.1, and $\varepsilon$ to 8/255 for the PGD attack. We also test other attack methods, such as AutoAttack and PatchFool (Fu et al., 2022) in Appendix G. We select 69 neural architectures trained on ImageNet (Russakovsky et al., 2015) from the PyTorch Image Models library (Wightman, 2019)[1]. Notice that to reduce the unexpected effect of a significant accuracy gap, the chosen model candidates are limited to the models whose top-1 accuracy is between 79% and 83%. We also ignore the models with unusual training techniques, such as training on extra training datasets, using a small or large model input resolution (*e.g.*, less than 200 or larger than 300), or knowledge distillation. When $A$ and $B$ take different input resolutions, then we resize the attacked image from the source network for the target network. We also sub-sample 10% ImageNet validation images (*i.e.*, 5,000 images) to measure the similarity. This strategy makes our similarity score more computationally efficient. The full pairwise similarities of 69 models can be found in Appendix D.

## 3 MODEL ANALYSIS BY NETWORK SIMILARITY

We analyze how each model component affects the diversity between networks quantitatively by feature importance analysis and clustering analysis (Sec. 3.1). We also show that the diversity caused by different training strategies is not as significant as caused by different neural architectures (Sec. 3.2).

### 3.1 WHICH ARCHITECTURAL COMPONENT CAUSES THE DIFFERENCE MOST?

**Settings.** We list 13 components of each architecture, *e.g.*, normalization (*e.g.*, BN (Ioffe & Szegedy, 2015) and LN (Ba et al., 2016)), activation functions (*e.g.*, ReLU (Krizhevsky et al., 2012) and GeLU (Ramachandran et al., 2017)) the existence of depthwise convolution, or stem layer (*e.g.*, 7×7 conv, 3×3 conv, or 16×16 conv with stride 16 – a.k.a. *"patchify"* stem (Liu et al., 2022)). The list of sub-modules is shown in Tab. 1 and Appendix B. We then convert each architecture as a feature vector based on the listed sub-modules. For example, we convert ResNet as $f_{\text{ResNet}} = [\text{Base arch} = \text{CNN}, \text{Norm} = \text{BN}, \text{Activation} = \text{ReLU}, \ldots]$.

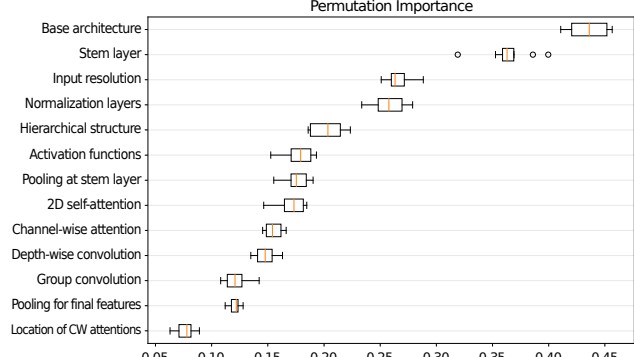

Figure 1: **Importance of architectural components to network similarity.** 13 components (Tab. 1) are sorted by the contribution to the network similarities. The larger feature importance means the component contributes more to the network similarity.

---

[1] https://github.com/rwightman/pytorch-image-models

**Feature important analysis.** The full list of components of 69 architectures is in Appendix B. Now, we measure the feature importance by fitting a gradient boosting regressor (Friedman, 2001) on the feature difference (*e.g.*, $f_{\text{ResNet-50}} - f_{\text{DeiT-base}}$) measured by Hamming distance and the corresponding similarity (*e.g.*, 2.3). The details of the regressor are in Appendix B. We use the permutation importance (Breiman, 2001); We measure how the trained regression model changes the predicted similarity by randomly changing each feature (*e.g.*, normalization). The feature importance of each architectural component is shown in Fig. 1. We first observe that the choice of base architecture (*e.g.*, CNN (Krizhevsky et al., 2012), Transformer (Vaswani et al., 2017), and MLP-Mixer (Tolstikhin et al., 2021)) contributes to the similarity most significantly. Fig. 1 also shows that the design choice of the input layer (*i.e.*, stem layer design choice or input resolution) affects the similarity as much as the choice of basic components such as normalization layers, activation functions, and the existence of attention layers. On the other hand, we observe that the modified efficiency-aware convolution operations, such as depth-wise convolution (Chollet, 2017) or group convolution (Xie et al., 2017), are not effective for model diversity.

**Clustering analysis.** We additionally provide a clustering analysis based on the architecture similarities. We construct a pairwise similarity graph $A$ between all 69 architectures where its vertex denotes an architecture, and its edge denotes the similarity between two networks. We perform the spectral clustering algorithm (Ng et al., 2001) on $A$ where the number of clusters $K$ is set to 10: We compute the Laplacian matrix of $A$, $L = D - A$ where $D$ is the diagonal matrix and its $i$-th component is $\sum_j A_{ij}$. Then, we perform K-means clustering on the $K$-largest eigenvectors of $L$, where $K$ is the number of clusters (*i.e.*, $K = 10$). More details of spectral clustering are in Appendix C.

Tab. 2 shows the clustering results on 69 networks and the top-5 keywords for each cluster based on term frequency-inverse document frequency (TF-IDF) analysis. Specifically, we treat each model feature (Tab. 1) as a word and compute TF and IDF by treating each architecture as a document. Then we compute the average TF-IDF for each cluster and report top-5 keywords. Similar to Fig. 1, the base architecture (*e.g.*, CNN in Cluster 5, 6, 10 and Transformer in Cluster 2, 3) and the design choice for the stem layer (*e.g.*, Cluster 1, 2, 4, 5, 6, 7, 8, 10) repeatedly appear at the top keywords. Especially, we can observe that the differences in base architecture significantly cause the diversity in model similarities, *e.g.*, non-hierarchical Transformers (Cluster 1), hierarchical networks with the patchified stem (Cluster 2), hierarchical Transformers (Cluster 3), CNNs with 2D self-attention (Cluster 4, 5), ResNet-based architectures (Cluster 6), and NAS-based architectures (Cluster 7).

## 3.2 TRAINING STRATEGY AND SIMILARITY SCORE

The architectural difference is not the only cause of the model diversity. We compare the effect by different architecture choices (*e.g.*, ResNet and Vision Transformer) and by different training strategies, while fixing the model architecture, as follows: **Different initializations** can affect the model training by the nature of the stochasticity of the training procedure. For example, Somepalli et al. (2022) showed that the decision boundary of each architecture can vary by different initializations. We also consider **different optimization hyper-parameters** (*e.g.*, learning rate, weight decay). Finally, we study the effect of **different training regimes** (*e.g.*, augmentations, type of supervision). For example, the choice of data augmentation (*e.g.*, Mixup (Zhang et al., 2018), or CutMix (Yun et al., 2019)) or the optimization techniques (*e.g.*, (Szegedy et al., 2016)) can theoretically or empirically affect adversarial robustness (Zhang et al., 2021; Park et al., 2022; Shafahi et al., 2019; Chun et al., 2019). Similarly, we investigate the effect of different types of supervision, such as self-supervision (*e.g.*, MOCO (Chen et al., 2021c), MAE (He et al., 2022) and BYOL (Grill et al., 2020)) or semi-weakly supervised learning (Yalniz et al., 2019).

**Settings. Different initializations:** We train 21 `ResNet-50` models and 16 `ViT-S` from scratch individually by initializing each network with different random seeds. **Different hyper-parameters:** We train 28 `ResNet-50` models and 9 `ViT-S` by choosing different learning rates, weight decays, and learning rate schedulers. The full list of hyper-parameters is shown in Appendix E. **Different training regimes:** We collect 23 `ResNet-50` models and 7 `ViT-S` models with different training

---

[†]Customized models by Wightman (2019): HaloRegNetZ = HaloNet + RegNetZ; ECA-BoTNeXt = ECA-Net + HaloNet + ResNeXt; ECA-BoTNeXt = ECA-Net + BoTNet + ResNeXt; LamHaloBoTNet = LambdaNet + HaloNet + BoTNet; SE-BoTNet = SENet + BoTNet; SE-HaloNet = SENet + HaloNet; Halo2BoTNet = HaloNet + BoTNet; NFNet-L0 = an efficient NFNet-F0 (Brock et al., 2021b); ECA-NFNet-L0 = ECA-Net + NFNet-L0; ResNet-V2-D-EVOS = ResNet-V2 + EvoNorms (Liu et al., 2020).

Table 2: **Clustering by architecture similarities.** We perform a clustering analysis on 69 neural architectures. All the architectures here are denoted by the aliases defined in their respective papers. We show the top-5 keywords for each cluster based on TF-IDF. InRes, SA, and CWA denote input resolution, self-attention, and channel-wise attention, respectively. The customized model details are described in the footnote[†].

| No. | Top-5 Keywords | Architecture |
|---|---|---|
| 1 | Stem layer: 16×16 conv w/ s16, No Hierarchical, GeLU, LN, Final GAP | ConViT-B (d'Ascoli et al., 2021), CrossViT-B (Chen et al., 2021a), DeiT-B (Touvron et al., 2021), DeiT-S (Touvron et al., 2021), ViT-S (patch size 16) (Dosovitskiy et al., 2021), ResMLP-S24 (Touvron et al., 2022a), gMLP-S (Liu et al., 2021a) |
| 2 | Stem layer: 4×4 conv w/ s4, LN, GeLU, Transformer, No pooling at stem | Twins-PCPVT-B (Chu et al., 2021), Twins-SVT-S (Chu et al., 2021), CoaT-Lite Small (Dai et al., 2021b), NesT-T (Zhang et al., 2022b), Swin-T (Liu et al., 2021b), S3 (Swin-T) (Chen et al., 2021b), ConvNeXt-T (Liu et al., 2022), ResMLP-B24 (Touvron et al., 2022a) |
| 3 | Transformer, Final GAP, GeLU, Pooling at stem, InRes: 224 | XCiT-M24 (Ali et al., 2021), XCiT-T12 (Ali et al., 2021), HaloRegNetZ-B[†], TNT-S (Han et al., 2021b), Visformer-S (Chen et al., 2021d), PiT-S (Heo et al., 2021), PiT-B (Heo et al., 2021) |
| 4 | Stem layer: stack of 3×3 conv, 2D SA, InRes: 256, Pooling at stem, SiLU | HaloNet-50 (Vaswani et al., 2021), LambdaResNet-50 (Bello, 2021), BoTNeT-26 (Srinivas et al., 2021), GC-ResNeXt-50 (Cao et al., 2019), ECAHaloNeXt-50[†], ECA-BoTNeXt-26[†] |
| 5 | Stem layer: stack of 3×3 convs, InRes: 256, 2D SA, CWA: middle of blocks, CNN | LamHaloBoTNet-50[†], SE-BoTNet-33[†], SE-HaloNet-33[†], Halo2BoTNet-50[†], GC-ResNet-50 (Cao et al., 2019), ECA-Net-33 (Wang et al., 2020c) |
| 6 | Stem layer: 7×7 conv w/ s2, ReLU, Pooling at stem, CNN, BN | ResNet-50 (He et al., 2016a), ResNet-101 (He et al., 2016a), ResNeXt-50 (Xie et al., 2017), Wide ResNet-50 (Zagoruyko & Komodakis, 2016), SE-ResNet-50 (Hu et al., 2018), SE-ResNeXt-50 (Hu et al., 2018), ResNet-V2-50 (He et al., 2016b), ResNet-V2-101 (He et al., 2016b), ResNet-50 (GN) (Wu & He, 2018), ResNet-50 (BlurPool) (Zhang, 2019), DPN-107 (Chen et al., 2017), Xception-65 (Chollet, 2017) |
| 7 | NAS, Stem layer: 3×3 conv w/ s2 CWA: middle of blocks, CWA, DW Conv | EfficientNet-B2 (Tan & Le, 2019a), FBNetV3-G (Dai et al., 2021a), ReXNet (×1.5) (Han et al., 2021a), RegNetY-32 (Radosavovic et al., 2020), MixNet-XL (Tan & Le, 2019b), NF-RegNet-B1 (Brock et al., 2021a) |
| 8 | Input resolution: 224, Stem layer: stack of 3×3 convs, Group Conv, Final GAP, 2D SA | NFNet-L0[†], ECA-NFNet-L0[†], PoolFormer-M48 (Yu et al., 2022), ResNeSt-50 (Zhang et al., 2022a), ResNet-V2-50-D-EVOS[†], ConvMixer-1536/20 (Trockman & Kolter, 2022) |
| 9 | ReLU, Input resolution: 224, DW Conv, BN 2D self-attention | ViT-B (patch size 32) (Dosovitskiy et al., 2021), R26+ViT-S (Steiner et al., 2022), DLA-X-102 (Yu et al., 2018), eSE-VoVNet-39 (Lee & Park, 2020), ResNet-101-C (He et al., 2019), RegNetX-320 (Radosavovic et al., 2020), HRNet-W32 (Wang et al., 2020b) |
| 10 | ReLU + Leaky ReLU, InRes: 256, Stem layer: 7×7 conv, CNN, Pooling at stem | CSPResNet-50 (Wang et al., 2020a), CSPResNeXt-50 (Wang et al., 2020a), CSPDarkNet-53 (Bochkovskiy et al., 2020), NF-ResNet-50 (Brock et al., 2021a) |

regimes. 22 ResNet-50 models include 1 model trained by PyTorch (Paszke et al., 2019), 4 models trained by GluonCV (Guo et al., 2020), a semi-supervised and semi-weakly supervised models on billion-scale unlabeled images (Yalniz et al., 2019), 5 models trained by different augmentations (Cutout (DeVries & Taylor, 2017), Mixup (Zhang et al., 2018), manifold Mixup (Verma et al., 2019), CutMix (Yun et al., 2019), and feature CutMix); 11 optimized models by Wightman et al. (2021). 7 ViT-S models contain the original ViT training setup (Dosovitskiy et al., 2021), a stronger data augmentation setup (Touvron et al., 2021), the training setup with distillation (Touvron et al., 2021), an improved DeiT training setup (Touvron et al., 2022b), and self-supervised training fashions including MoCo v3 (Chen et al., 2021c), MAE (He et al., 2022) and BYOL (Grill et al., 2020).

Tab. D.1 shows the comparison of similarity scores between the same architecture but different learning methods (a smaller similarity means more diversity). First, we observe that using different random initialization or different optimization hyper-parameters shows high correlations with each other (almost $\geq 4.2$) while the average similarity score between various neural architectures is 2.73. In other words, the difference in initializations or optimization hyper-parameters does not significantly contribute to the model diversity. Second, we observe that using different learning technique (*e.g.*, different augmentation methods, and different types of supervision) remarkably affect the similarity score (3.27 for ResNet and 3.44 for ViT), but is not as significant as the architectural difference (2.73). Furthermore, we discover that the change of similarity caused by different initializations or hyper-parameters is less marked than the change caused by different architecture (Fig. D.1).

In practice, learning the same network with significantly different learning techniques is challenging and time-consuming because the hyper-parameter search space is tremendously large (Bergstra & Bengio, 2012). Especially, a novel optimization technique (*e.g.*, using more data points for semi- or self-supervised learning, using different augmentation methods, or adopting advanced optimization settings) sometimes needs a completely different optimization setting compared to the vanilla setting. On the other hand, learning a new architecture is easier than using a different learning paradigm in terms of hyper-parameter selection and implementation. Thus, we can conclude that if we need multiple diverse models, then using multiple architectures is more effective and easier to optimize.

## 4 PRACTICAL APPLICATIONS OF THE NETWORK SIMILARITY SCORE

In this section, we show the practical usage of our similarity score. First, we show that using more diverse models will lead to better ensemble performance (Sec. 4.1). Second, we suggest a similarity-based guideline for the choice of a teacher model when distilling to a specific architecture. Different from several existing studies that show a more similar teacher is better, our findings show that their observation is valid only if the teacher and student networks are based on the same architecture (*e.g.*, Transformer). On the other hand, when the teacher and student networks are based on different architectures, a more dissimilar teacher is better (Sec. 4.2).

### 4.1 MODEL DIVERSITY AND ENSEMBLE PERFORMANCE

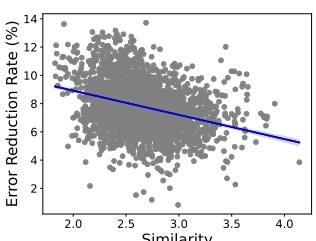

The model ensemble is a practical technique to achieve high performances in practice, however, there is still not enough study on the choice of the models for the ensemble in large-scale complex models. Previous studies are mainly conducted on very small datasets and linear models (Kuncheva & Whitaker, 2003). We report the relationship between the model similarity and the ensemble performance based on the unweighted average method (Ju et al., 2018) (*i.e.*, averaging the logit values of the ensembled models). Because the ensemble performance is sensitive to the original model performances, we define Error Reduction Rate (ERR) as $1 - \frac{\mathrm{Err_{ens}}(M)}{\frac{1}{|M|}\sum_{m \in M}\mathrm{Err}(m)}$, where $M$ is the set of the ensembled models, $\mathrm{ERR}(m)$ denotes the top-1 ImageNet validation error of model $m$, and $\mathrm{Err_{ens}}(\cdot)$ denotes the top-1 error of the ensemble.

Figure 2: **Correlation between model similarity and ensemble performance.** The trend line and its 90% confidence interval are shown in the blue line.

We first measure the 2-ensemble performances among the 69 neural architectures in Tab. 2 (*i.e.*, the number of ensembles is $\binom{69}{2} = 2346$). We plot the relationship between the model similarity and ERR in Fig. 2. We observe that there exists a strong negative correlation between the model similarity and the ensemble performance (Pearson correlation coefficient $-0.31$ with p-value $1.74 \times 10^{-54}$), *i.e.*, *more diversity leads to better ensemble performance*.

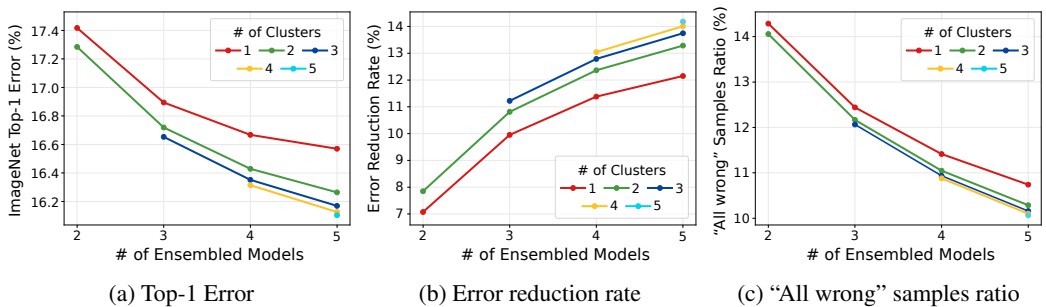

| (a) Top-1 Error | (b) Error reduction rate | (c) "All wrong" samples ratio |
| --- | --- | --- |

Figure 3: **Model diversity by architecture difference and ensemble performance.** We report ensemble performances by varying the number of ensembled models ($N$) and the diversity of the ensembled models. The diversity is controlled by choosing the models from $k$ different clusters (larger $k$ denotes more diversity).

We also conduct $N$-ensemble experiments with $N \geq 2$ based on the clustering results in Tab. 2. We evaluate the average ERR of the ensemble of models from $k$ clusters, *i.e.*, if $N = 5$ and $k = 3$, the ensembled models are only sampled from the selected 3 clusters while ignoring the other 7 clusters. We investigate the effect of model diversity and ensemble performance by examining $k = 1 \ldots N$ (*i.e.*, larger $k$ denotes more diverse ensembled models). We report three evaluation metrics in Fig. 3: ImageNet top-1 error of the ensemble (lower is better), error reduction rate (ERR) (higher is better), and the number of wrong samples by all ensembled models (lower is better). In all metrics, we observe that the ensemble of more diverse models (*i.e.*, by increasing the number of clusters for the model selection) shows better ensemble performance. Interestingly, in Fig. 3a, we observe that when the number of clusters for the model selection ($k$) is decreased, the ensemble performance by the number of ensembled models ($N$) quickly reaches saturation. Similarly, in Fig. 3c, we observe that the number of wrong samples by all models is decreased by selecting more diverse models. In other words, the different architecture choices can lead to different model decision boundaries.

In Sec. 3.2, we showed that the different training strategies are not as effective as different architectures in terms of diversity. To examine the effect of this observation on the ensemble scenario, we report the ensemble results of different training strategies, *i.e.*, the same `ResNet-50` and `ViT-S` models in Tab. D.1. For comparison with different architectures, we also report the ensemble of different architectures where all ensembled models are from different clusters (*i.e.*, when $N = k$ in Fig. 3). Fig. 4 shows that although using diverse training regimes (blue lines) improves ensemble performance most among other training techniques (red and green lines), the improvements by using different architectures (yellow lines) are more significant than the improvements by using different training regimes (blue lines) with large gaps.

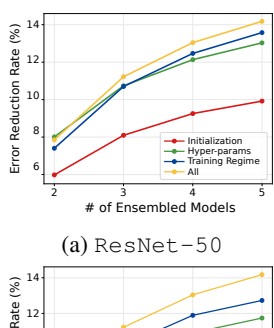

(a) `ResNet-50`

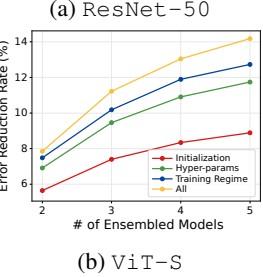

(b) `ViT-S`

Figure 4: **Model diversity by training techniques and ensemble performance.** The same metrics to Fig. 3 for `ResNet-50` and `ViT-S` models in Tab. D.1 are shown. "All" means $N = k$ in Fig. 3 (*i.e.*, # of ensembled models = # of Clusters).

## 4.2 MODEL DIVERSITY AND KNOWLEDGE DISTILLATION

Knowledge distillation (Hinton et al., 2015) is a training method for transferring rich knowledge of a well-trained teacher network. Intuitively, distillation performance affects a lot by choice of the teacher network, however, the relationship between similarity and distillation performance has not yet been explored enough, especially for ViT. In this subsection, we investigate how the similarity between teacher and student networks contributes to the distillation performance. There are several studies showing two contradictory conclusions; Jin et al. (2019); Mirzadeh et al. (2020) showed that a similar teacher leads to better distillation performance; Touvron et al. (2021) reports that distillation from a substantially different teacher is beneficial for ViT training.

**Settings.** We use the same distillation setting by Touvron et al. (2021); We train `ViT-Ti` models with the hard distillation strategy (Hinton et al., 2015) (*i.e.*, using the predicted label by the teacher as the ground truth label). We train 25 `ViT-Ti` models with different teacher networks from Tab. 2. Here, we include all non-ViT teachers that have a similarity score larger than 2.5, and 25 models among 67 models (except for ViT and DeiT) are sampled according to the distribution of the similarity scores to prevent sampling bias. We describe the detailed training settings in Appendix F.1.

Fig. 5a illustrates the relationship between the teacher-student similarity and the distillation performance. Fig. 5a tends to show a not significant negative correlation between teacher-student similarity and distillation performance ($-0.32$ Pearson correlation coefficient with 0.12 p-value). However, if we only focus on when the teacher and student networks are based on the same architecture (*i.e.*, Transformer), we can observe a strong positive correlation (Fig. 5b) – 0.70 Pearson correlation coefficient with 0.078 p-value. In this case, our observation is aligned with Jin et al. (2019); Mirzadeh et al. (2020): a teacher similar to the student improves distillation performance. However, when the teacher and student networks are based on different architectures (*e.g.*, CNN), then we can observe a stronger negative correlation (Fig. 5c) with $-0.51$ Pearson correlation coefficient and 0.030 p-value. In this case, a more dissimilar teacher leads to better distillation performance. We also test

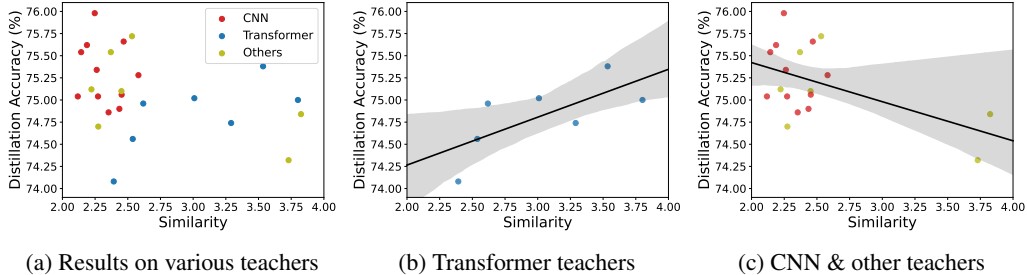

(a) Results on various teachers      (b) Transformer teachers      (c) CNN & other teachers

Figure 5: **Model diversity and distillation performance.** (a) We show the relationship between teacher-student similarity and distillation performance of 25 `DeiT-S` models distilled by various teacher networks. We show the relationship when the teacher and student networks are based on (b) Transformer and (c) otherwise.

other factors that can affect distillation performance in Appendix F.2; We observe that distillation performance is not correlated to teacher accuracy in our experiments.

Why we observe contradictory results for Transformer teachers (Fig. 5b) and other teachers (Fig. 5c)? Here, we conjecture that when the teacher and student networks differ significantly, distillation works as a strong regularizer. In this case, using a more dissimilar teacher can be considered as a stronger regularizer (Fig. 5c). On the other hand, we conjecture that if two networks are similar, then distillation works as easy-to-follow supervision for the student network. In this case, we can assume that a more similar teacher will work better because a more similar teacher will provide more easy-to-follow supervision for the student network (Fig. 5b). In our experiments, we found that the regularization effect improves distillation performance better than easy-to-follow supervision (*i.e.*, the best performing distillation result is by a CNN teacher). Therefore, in practice, we recommend using a significantly different teacher network for achieving better distillation performance (*e.g.*, using RegNet (Radosavovic et al., 2020) teacher for ViT student as Touvron et al. (2021)).

## 5 LIMITATIONS AND DISCUSSIONS

We include three discussions related to our similarity score. Due to the space limitation, the full details of each item can be found in Appendix G. Our discussions include:

- Robustness of our similarity score to the choice of adversarial attack method. We show that the similarity rankings by PGD (Madry et al., 2018), AutoAttack (Croce & Hein, 2020), and PatchFool (Fu et al., 2022) do not differ significantly.
- How adversarially trained models affect our similarity scores.
- An efficient approximation of our analysis for a novel model (*i.e.*, for a novel 70th model for our analysis). We show that instead of generating adversarial attacks for all 70 models, just using the attacked images from the novel model could be enough.
- Other possible applications of our similarity score (*e.g.*, a novel architecture development, analyzing the hyper-parameter sensitivity, model selection for selecting diverse models).

## 6 CONCLUSION

We have explored similarities between image classification models to investigate what makes the model diverse and whether developing and using diverse models are required. For quantitative and model-agnostic assessment of the similarity, we have suggested a new score based on attack transferability demonstrating differences in input gradients. Using our new similarity function, we conduct a large-scale and extensive analysis using 69 state-of-the-art ImageNet models. We have shown that macroscopic architectural properties, such as base architecture and stem architecture, have a greater impact on similarity than microscopic operations, such as types of used convolution, with numerical analysis. Furthermore, the effectiveness of the training strategy is minor compared to model architectural design-related features. Finally, we have firstly discovered the advantages of using diverse models in ensemble or distillation strategies for a practical scenario with a large-scale training dataset and a highly complex architecture.

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

APPENDIX

We include additional materials in this document, such as the details of our similarity function (A) the network details (B), spectral clustering method (C), training settings (E), setting of knowledge distillation (F.1), and the reason of using of attack transferability for comparing decision boundary (H). We also provide experimental results of computing pairwise similarities (D) and distillation performance (F.2). We discuss some limitations and discussion points of our work (G), such as the effect of the choice of attack methods, an efficient approximation, and possible applications.

## A  MORE DETAILS OF OUR SIMILARITY FUNCTION

Ideally, we would like to define a similarity function between two models with the following three properties: (1) $x = \arg\min_y d(x, y)$ (2) $d(x, y) = d(y, x)$ (3) $d(x, y) > d(x, x)$ if $x \neq y$. The adversarial attack transferability generally satisfies these properties. If we assume the adversary is optimal, then the transferred attacked accuracy ($\text{acc}_{B \rightarrow A}$) will be zero and it will be the minimum (because accuracy is non-negative). $\text{acc}_{A \rightarrow B} + \text{acc}_{B \rightarrow A}$ is symmetric. It will satisfy $d(x, y) \geq d(x, x)$ if $x \neq y$ where it is a weaker condition than (3). In other words, adversarial attack transferability and equation 1 can serve as a similarity score function from a functional point of view.

## B  DETAILS OF NETWORKS USED FOR OUR ANALYSIS

We use 69 models in our research to evaluate the similarity between models and to investigate the impact of model diversity. In the main paper, we mark the names of models based on their research paper and PyTorch Image Models library (`timm`; 0.6.7 version) (Wightman, 2019). Tab. B.1 shows the full list of the models based on their research paper and `timm` alias.

In Tab. B.2 and Tab. B.3, We show the full network specification defined by Tab. 1. We follow the corresponding paper and `timm` library to list the model features.

We fit a gradient boosting regressor (Friedman, 2001) based on `Scikit-learn` (Pedregosa et al., 2011) and report the permutation importance of each component in Fig. 1. The number of boosting stages, maximum depth, minimum number of samples, and learning rate are set to 500, 12, 4, and 0.02, respectively. Permutation importance is computed by permuting a feature 10 times.

Table B.1: Lists of 69 models and their names based on their research paper and `timm` library.

| in timm | in paper | in timm | in paper | in timm | in paper |
|---|---|---|---|---|---|
| botnet26t_256 | BoTNet-26 | gluon_xception65 | Xception-65 | resnet50_gn | ResNet-50 (GN) |
| coat_lite_small | CoaT-Lite Small | gmlp_s16_224 | gMLP-S | resnetblur50 | ResNet-50 (BlurPool) |
| convit_base | ConViT-B | halo2botnet50ts_256 | Halo2BoTNet-50 | resnetv2_101 | ResNet-V2-101 |
| convmixer_1536_20 | ConvMixer-1536/20 | halonet50ts | HaloNet-50 | resnetv2_50 | ResNet-V2-50 |
| convnext_tiny | ConvNeXt-T | haloregnetz_b | HaloRegNetZ | resnetv2_50d_evos | ResNet-V2-50-EVOS |
| crossvit_base_240 | CrossViT-B | hrnet_w64 | HRNet-W32 | resnext50_32x4d | ResNeXt-50 |
| cspdarknet53 | CSPDarkNet-53 | jx_nest_tiny | NesT-T | rexnet_150 | ReXNet (×1.5) |
| cspresnet50 | CSPResNet-50 | lambda_resnet50ts | LambdaResNet-50 | sebotnet33ts_256 | SEBoTNet-33 |
| cspresnext50 | CSPResNeXt-50 | lamhalobotnet50ts_256 | LamHaloBoTNet-50 | sehalonet33ts | SEHaloNet-33 |
| deit_base_patch16_224 | DeiT-B | mixnet_xl | MixNet-XL | seresnet50 | SEResNet-50 |
| deit_small_patch16_224 | DeiT-S | nf_regnet_b1 | NF-RegNet-B1 | seresnext50_32x4d | SEResNeXt-50 |
| dla102x2 | DLA-X-102 | nf_resnet50 | NF-ResNet-50 | swin_s3_tiny_224 | S3 (Swin-T) |
| dpn107 | DPN-107 | nfnet_l0 | NFNet-L0 | swin_tiny_patch4_window7_224 | Swin-T |
| eca_botnet26ts_256 | ECA-BoTNeXt-26 | pit_b_224 | PiT-B | tnt_s_patch16_224 | TNT-S |
| eca_halonext26ts | ECA-HaloNeXt-26 | pit_s_224 | PiT-S | twins_pcpvt_base | Twins-PCPVT-B |
| eca_nfnet_l0 | ECA-NFNet-L0 | poolformer_m48 | PoolFormer-M48 | twins_svt_small | Twins-SVT-S |
| eca_resnet33ts | ECA-ResNet-33 | regnetx_320 | RegNetX-320 | visformer_small | VisFormer-S |
| efficientnet_b2 | EfficientNet-B2 | regnety_032 | RegNetY-32 | vit_base_patch32_224 | ViT-B |
| ese_vovnet39b | eSE-VoVNet-39 | resmlp_24_224 | ResMLP-S24 | vit_small_patch16_224 | ViT-S |
| fbnetv3_g | FBNetV3-G | resmlp_big_24_224 | ResMLP-B24 | vit_small_r26_s32_224 | R26+ViT-S |
| gcresnet50t | GCResNet-50 | resnest50d | ResNeSt-50 | wide_resnet50_2 | Wide ResNet-50 |
| gcresnext50ts | GCResNeXt-50 | resnet101 | ResNet-101 | xcit_medium_24_p16_224 | XCiT-M24 |
| gluon_resnet101_v1c | ResNet-101-C | resnet50 | ResNet-50 | xcit_tiny_12_p8_224 | XCiT-T12 |

## C  SPECTRAL CLUSTERING DETAILS

We use the normalized Laplacian matrix to compute the Laplacian matrix. We also run K-means clustering 100 times and choose the clustering result with the best final objective function to reduce the randomness by the K-means clustering algorithm.

Table B.2: Description of features of 69 models. "s" in "Stem layer" indicates the stride of a layer in the stem, and the number before and after "s" are a kernel size and size of stride, respectively. For example, "3s2/3/3" means that the stem is composed of the first layer having 3 × 3 kernel with stride 2, the second layer having 3 × 3 kernel with stride 1, and the last layer having 3 × 3 with stride 1.

| Model name | Base architecture | Hierarchical structure | Stem layer | Input resolution | Normalization | Activation |
|---|---|---|---|---|---|---|
| botnet26t_256 | CNN | Yes | 3s2/3/3 | 256 × 256 | BN | ReLU |
| convmixer_1536_20 | CNN | Yes | 7s7 | 224 × 224 | BN | GeLU |
| convnext_tiny | CNN | Yes | 4s4 | 224 × 224 | LN | GeLU |
| cspdarknet53 | CNN | Yes | 3s1 | 256 × 256 | BN | Leaky ReLU |
| cspresnet50 | CNN | Yes | 7s2 | 256 × 256 | BN | Leaky ReLU |
| cspresnext50 | CNN | Yes | 7s2 | 256 × 256 | BN | Leaky ReLU |
| dla102x2 | CNN | Yes | 7s1 | 224 × 224 | BN | ReLU |
| dpn107 | CNN | Yes | 7s2 | 224 × 224 | BN | ReLU |
| eca_botnext26ts_256 | CNN | Yes | 3s2/3/3 | 256 × 256 | BN | SiLU |
| eca_halonext26ts | CNN | Yes | 3s2/3/3 | 256 × 256 | BN | SiLU |
| eca_nfnet_l0 | CNN | Yes | 3s2/3/3/3s2 | 224 × 224 | Norm-free | SiLU |
| eca_resnet33ts | CNN | Yes | 3s2/3/3s2 | 256 × 256 | BN | SiLU |
| ese_vovnet39b | CNN | Yes | 3s2/3/3s2 | 224 × 224 | BN | ReLU |
| gcresnet50t | CNN | Yes | 3s2/3/3s2 | 256 × 256 | LN + BN | ReLU |
| gcresnext50ts | CNN | Yes | 3s2/3/3 | 256 × 256 | LN + BN | ReLU + SiLU |
| gluon_resnet101_v1c | CNN | Yes | 3s2/3/3 | 224 × 224 | BN | ReLU |
| gluon_xception65 | CNN | Yes | 3s2/3 | 299 × 299 | BN | ReLU |
| halo2botnet50ts_256 | CNN | Yes | 3s2/3/3s2 | 256 × 256 | BN | SiLU |
| halonet50ts | CNN | Yes | 3s2/3/3 | 256 × 256 | BN | SiLU |
| hrnet_w64 | CNN | Yes | 3s2/3s2 | 224 × 224 | BN | ReLU |
| lambda_resnet50ts | CNN | Yes | 3s2/3/3 | 256 × 256 | BN | SiLU |
| lamhalobotnet50ts_256 | CNN | Yes | 3s2/3/3s2 | 256 × 256 | BN | SiLU |
| nf_resnet50 | CNN | Yes | 7s2 | 256 × 256 | Norm-free | ReLU |
| nfnet_l0 | CNN | Yes | 3s2/3/3/3s2 | 224 × 224 | Norm-free | ReLU + SiLU |
| poolformer_m48 | CNN | Yes | 7s4 | 224 × 224 | GN | GeLU |
| resnest50d | CNN | Yes | 3s2/3/3 | 224 × 224 | BN | ReLU |
| resnet101 | CNN | Yes | 7s2 | 224 × 224 | BN | ReLU |
| resnet50 | CNN | Yes | 7s2 | 224 × 224 | BN | ReLU |
| resnet50_gn | CNN | Yes | 7s2 | 224 × 224 | GN | ReLU |
| resnetblur50 | CNN | Yes | 7s2 | 224 × 224 | BN | ReLU |
| resnetv2_101 | CNN | Yes | 7s2 | 224 × 224 | BN | ReLU |
| resnetv2_50 | CNN | Yes | 7s2 | 224 × 224 | BN | ReLU |
| resnetv2_50d_evos | CNN | Yes | 3s2/3/3 | 224 × 224 | EvoNorm | - |
| resnext50_32x4d | CNN | Yes | 7s2 | 224 × 224 | BN | ReLU |
| sebotnet33ts_256 | CNN | Yes | 3s2/3/3s2 | 256 × 256 | BN | ReLU + SiLU |
| sehalonet33ts | CNN | Yes | 3s2/3/3s2 | 256 × 256 | BN | ReLU + SiLU |
| seresnet50 | CNN | Yes | 7s2 | 224 × 224 | BN | ReLU |
| seresnext50_32x4d | CNN | Yes | 7s2 | 224 × 224 | BN | ReLU |
| wide_resnet50_2 | CNN | Yes | 7s2 | 224 × 224 | BN | ReLU |
| convit_base | Transformer | No | 16s16 | 224 × 224 | LN | GeLU |
| crossvit_base_240 | Transformer | Yes | 16s16 | 240 × 240 | LN | GeLU |
| deit_base_patch16_224 | Transformer | No | 16s16 | 224 × 224 | LN | GeLU |
| deit_small_patch16_224 | Transformer | No | 16s16 | 224 × 224 | LN | GeLU |
| jx_nest_tiny | Transformer | Yes | 4s4 | 224 × 224 | LN | GeLU |
| pit_s_224 | Transformer | Yes | 16s8 | 224 × 224 | LN | GeLU |
| swin_tiny_patch4_window7_224 | Transformer | Yes | 4s4 | 224 × 224 | LN | GeLU |
| tnt_s_patch16_224 | Transformer | Yes | 7s4 | 224 × 224 | LN | GeLU |
| vit_base_patch32_224 | Transformer | No | 32s32 | 224 × 224 | LN | GeLU |
| vit_small_patch16_224 | Transformer | No | 16s16 | 224 × 224 | LN | GeLU |
| gmlp_s16_224 | MLP-Mixer | Yes | 16s16 | 224 × 224 | LN | GeLU |
| resmlp_24_224 | MLP-Mixer | No | 16s16 | 224 × 224 | Affine transform | GeLU |
| resmlp_big_24_224 | MLP-Mixer | Yes | 8s8 | 224 × 224 | Affine transform | GeLU |
| swin_s3_tiny_224 | NAS (TFM) | Yes | 4s4 | 224 × 224 | LN | GeLU |
| efficientnet_b2 | NAS (CNN) | Yes | 3s2 | 256 × 256 | BN | SiLU |
| fbnetv3_g | NAS (CNN) | Yes | 3s2 | 240 × 240 | BN | HardSwish |
| haloregnetz_b | NAS (CNN) | Yes | 3s2 | 224 × 224 | BN | ReLU + SiLU |
| mixnet_xl | NAS (CNN) | Yes | 3s2 | 224 × 224 | BN | ReLU + SiLU |
| nf_regnet_b1 | NAS (CNN) | Yes | 3s2 | 256 × 256 | Norm-free | ReLU + SiLU |
| regnetx_320 | NAS (CNN) | Yes | 3s2 | 224 × 224 | BN | ReLU |
| regnety_032 | NAS (CNN) | Yes | 3s2 | 224 × 224 | BN | ReLU |
| rexnet_150 | NAS (CNN) | Yes | 3s2 | 224 × 224 | BN | ReLU + SiLU + ReLU6 |
| coat_lite_small | Hybrid | Yes | 4s4 | 224 × 224 | LN | GeLU |
| pit_b_224 | Hybrid | Yes | 14s7 | 224 × 224 | LN | GeLU |
| twins_pcpvt_base | Hybrid | Yes | 4s4 | 224 × 224 | LN | GeLU |
| twins_svt_small | Hybrid | Yes | 4s4 | 224 × 224 | LN | GeLU |
| visformer_small | Hybrid | Yes | 7s2 | 224 × 224 | BN | GeLU + ReLU |
| vit_small_r26_s32_224 | Hybrid | No | 7s2 | 224 × 224 | LN + GN | GeLU + ReLU |
| xcit_medium_24_p16_224 | Hybrid | No | 3s2/3s2/3s2/3s2 | 224 × 224 | LN + BN | GeLU |
| xcit_tiny_12_p8_224 | Hybrid | No | 3s2/3s2/3s2 | 224 × 224 | LN + BN | GeLU |

Table B.3: Description of features of 69 models. "Pooling (stem)" and "Pooling (final)" denote "Pooling at the stem" and "Pooling for final feature", respectively. "SA", "CW", and "DW" means "Self-attention", "Channel-wise", and "Depth-wise", respectively.

| Model name | Pooling (stem) | Pooling (final) | 2D SA | CW attention | Location of CW attention | DW conv | Group conv |
|---|---|---|---|---|---|---|---|
| botnet26t_256 | Yes | GAP | Yes (BoT) | No | | No | No |
| convmixer_1536_20 | No | GAP | No | No | | Yes | No |
| convnext_tiny | No | GAP | No | No | | Yes | No |
| cspdarknet53 | No | GAP | No | No | | No | No |
| cspresnet50 | Yes | GAP | No | No | | No | No |
| cspresnext50 | Yes | GAP | No | No | | No | Yes |
| dla102x2 | No | GAP | No | No | | No | Yes |
| dpn107 | Yes | GAP | No | No | | No | Yes |
| eca_botnext26ts_256 | Yes | GAP | Yes (BoT) | Yes (ECA) | Middle | No | Yes |
| eca_halonext26ts | Yes | GAP | Yes (Halo) | Yes (ECA) | Middle | No | Yes |
| eca_nfnet_l0 | No | GAP | No | Yes (ECA) | End | No | Yes |
| eca_resnet33ts | No | GAP | No | Yes (ECA) | Middle | No | No |
| ese_vovnet39b | No | GAP | No | Yes (ESE) | End | No | No |
| gcresnet50t | No | GAP | No | Yes (GCA) | Middle | Yes | No |
| gcresnext50ts | Yes | GAP | No | Yes (GCA) | Middle | Yes | Yes |
| gluon_resnet101_v1c | Yes | GAP | No | No | | No | No |
| gluon_xception65 | No | GAP | No | No | | Yes | No |
| halo2botnet50ts_256 | No | GAP | Yes (Halo, BoT) | No | | No | No |
| halonet50ts | Yes | GAP | Yes (Halo) | No | | No | No |
| hrnet_w64 | No | GAP | No | No | | No | No |
| lambda_resnet50ts | Yes | GAP | Yes (Lambda) | No | | No | No |
| lamhalobotnet50ts_256 | No | GAP | Yes (Lambda, Halo, BoT) | No | | No | No |
| nf_resnet50 | Yes | GAP | No | No | | No | No |
| nfnet_l0 | No | GAP | No | Yes (SE) | End | No | Yes |
| poolformer_m48 | No | GAP | No | No | | No | No |
| resnest50d | Yes | GAP | No | Yes | | No | Yes |
| resnet101 | Yes | GAP | No | No | | No | No |
| resnet50 | Yes | GAP | No | No | | No | No |
| resnet50_gn | Yes | GAP | No | No | | No | No |
| resnetblur50 | Yes | GAP | No | No | | No | No |
| resnetv2_101 | Yes | GAP | No | No | | No | No |
| resnetv2_50 | Yes | GAP | No | No | | No | No |
| resnetv2_50d_evos | Yes | GAP | No | No | | No | No |
| resnext50_32x4d | Yes | GAP | No | No | | No | Yes |
| sebotnet33ts_256 | No | GAP | Yes (BoT) | Yes (SE) | Middle | No | No |
| sehalonet33ts | No | GAP | Yes (Halo) | Yes (SE) | Middle | No | No |
| seresnet50 | Yes | GAP | No | Yes (SE) | End | No | No |
| seresnext50_32x4d | Yes | GAP | No | Yes (SE) | End | No | Yes |
| wide_resnet50_2 | Yes | GAP | No | No | | No | No |
| convit_base | No | CLS token | No | No | | No | No |
| crossvit_base_240 | No | CLS token | No | No | | No | No |
| deit_base_patch16_224 | No | CLS token | No | No | | No | No |
| deit_small_patch16_224 | No | CLS token | No | No | | No | No |
| jx_nest_tiny | No | GAP | No | No | | No | No |
| pit_s_224 | No | CLS token | No | No | | Yes | No |
| swin_tiny_patch4_window7_224 | No | GAP | No | No | | No | No |
| tnt_s_patch16_224 | No | CLS token | No | No | | No | No |
| vit_base_patch32_224 | No | CLS token | No | No | | No | No |
| vit_small_patch16_224 | No | CLS token | No | No | | No | No |
| gmlp_s16_224 | No | GAP | No | No | | No | No |
| resmlp_24_224 | No | GAP | No | No | | No | No |
| resmlp_big_24_224 | No | GAP | No | No | | No | No |
| swin_s3_tiny_224 | No | GAP | No | No | | No | No |
| efficientnet_b2 | No | GAP | No | Yes (SE) | Middle | Yes | No |
| fbnetv3_g | No | GAP | No | Yes (SE) | Middle | Yes | No |
| haloregnetz_b | No | GAP | Yes (Halo) | Yes (SE) | Middle | No | Yes |
| mixnet_xl | No | GAP | No | Yes (SE) | Middle | Yes | No |
| nf_regnet_b1 | No | GAP | No | Yes (SE) | Middle | Yes | Yes |
| regnetx_320 | No | GAP | No | No | | No | Yes |
| regnety_032 | No | GAP | No | Yes (SE) | Middle | No | Yes |
| rexnet_150 | No | GAP | No | Yes (SE) | Middle | Yes | No |
| coat_lite_small | No | CLS token | No | No | | Yes | No |
| pit_b_224 | No | CLS token | No | No | | Yes | No |
| twins_pcpvt_base | No | GAP | No | No | | Yes | No |
| twins_svt_small | No | GAP | No | No | | Yes | No |
| visformer_small | No | GAP | No | No | | No | Yes |
| vit_small_r26_s32_224 | Yes | CLS token | No | No | | No | No |
| xcit_medium_24_p16_224 | No | CLS token | No | No | | Yes | No |
| xcit_tiny_12_p8_224 | No | CLS token | No | No | | Yes | No |

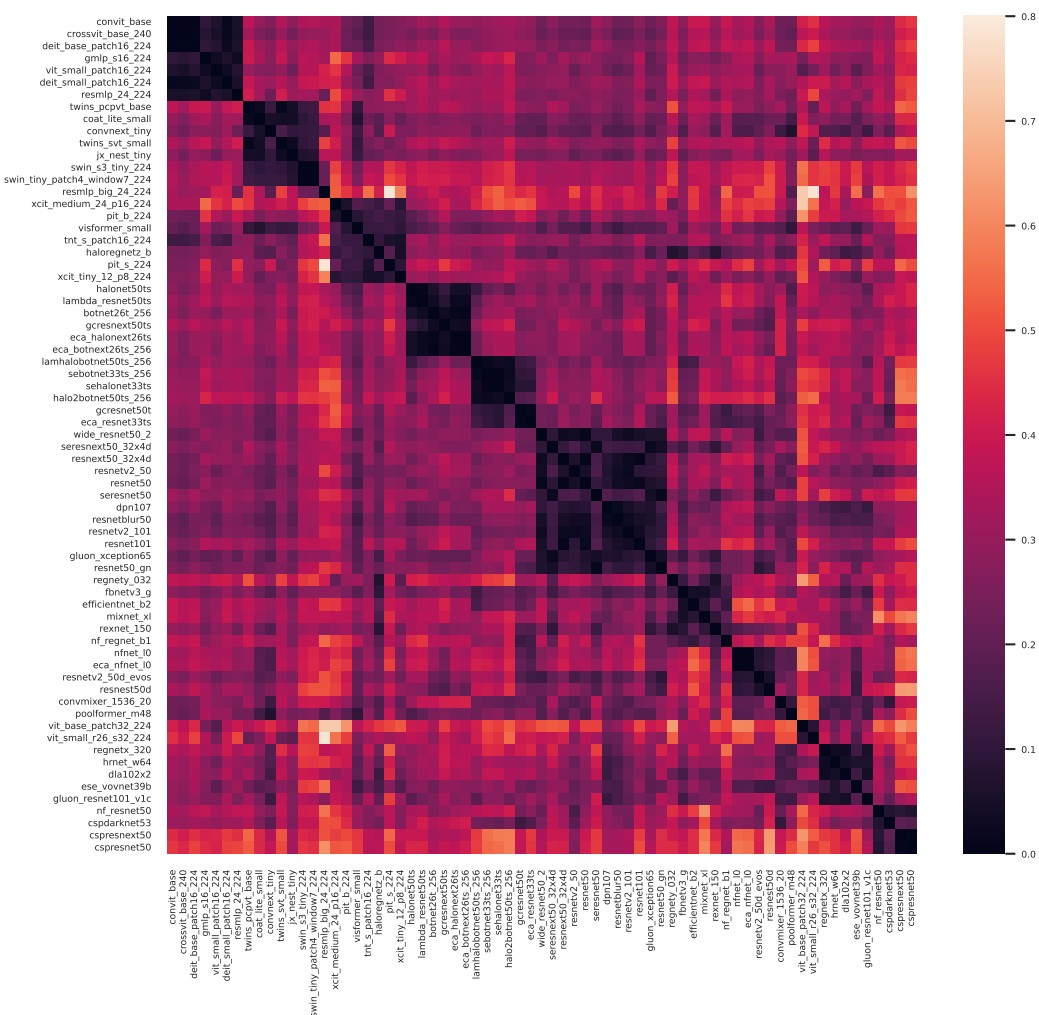

Figure C.1: **Spectral features of 69 architectures.** The $K$-th largest eigenvectors of the Laplacian matrix of the pairwise similarity graph of 69 architectures are shown ($K = 10$ in this figure). Rows and columns are sorted by the clustering index in Tab. 2. We denote the model name in `timm` for each row and column.

We visualize the pairwise distances of the spectral features (*i.e.*, $K$-largest eigenvectors of $L$) of 69 architectures in Fig. C.1. Note that rows and columns of Fig. C.1 are sorted by the clustering results. Fig. C.1 shows block diagonal patterns, *i.e.*, in-cluster similarities are large while between-cluster similarities are small.

## D   PAIRWISE SIMILARITIES

Fig. D.1 shows that the similarities between different architectures (Fig. D.1a and the off-diagonal elements of Fig. D.1b) are smaller than the similarities within the same architecture with different initializations or hyper-parameters.

Fig. D.2 indicates the pairwise similarity among 69 models. We can observe a weak block pattern around clusters, as also revealed in Fig. C.1.

Table D.1: **Similarity within the same architecture.** We compare the average similarity within the same network architecture but trained with different procedures, *e.g.*, different random initializations, different optimization hyper-parameters, and different training regimes. We report the similarities of two neural architectures, `ResNet-50` and `ViT-S`, where we use the DeiT training setting (Touvron et al., 2021) for ViT training. "All" denotes the average similarity of 69 neural architectures in Tab. 2.

| Architecture | Diversity by | Similarity |
|---|---|---|
| `ResNet-50` | Initialization | 4.23 |
| | Hyper-parameter | 4.05 |
| | Training regime | 3.27 |
| `ViT-S` | Initialization | 4.21 |
| | Hyper-parameter | 4.22 |
| | Training regime | 3.44 |
| | All | 2.73 |

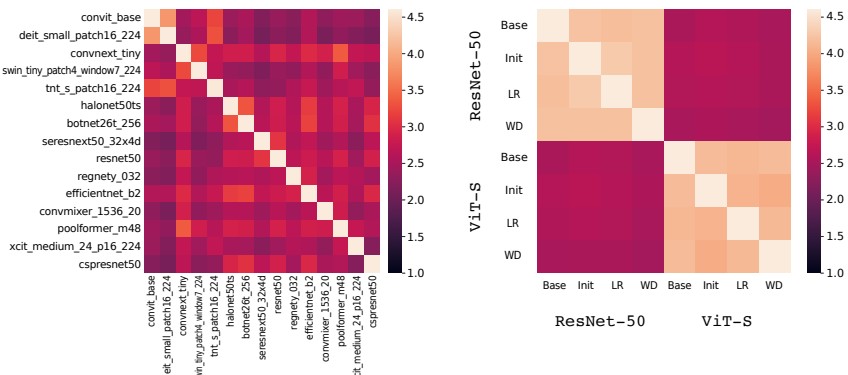

(a) Similarity between different networks. (b) Similarity within `ResNet-50` and `ViT-S` with different optimization settings.

Figure D.1: **Similarity scores between different architectures vs. within the same architecture.** (a) Subsampled pairwise similarity scores of 69 neural architectures. (b) Similarity between 4 `ResNet-50` models and 4 `ViT-S` models with different optimization settings. `Init`, `LR`, and `WD` are randomly chosen from models trained with different settings of initialization, learning rate, and weight decay in Tab. D.1, respectively.

# E  TRAINING SETTING DETAILS FOR SEC. 3.2

We train 21 `ResNet-50` models and 16 `ViT-S` from scratch individually by initializing each network with different random seeds. We further train 28 `ResNet-50` models by randomly choosing learning rate (×0.1, ×0.2, ×0.5, ×1, ×2, and ×5 where the base learning rate is 0.1), weight decay (×0.1, ×0.2, ×0.5, ×1, ×2, and ×5 where the base weight decay is 1e-4), and learning rate scheduler (step decay or cosine decay). Similarly, we train 9 `ViT-S` models by randomly choosing learning rate (×0.2, ×0.4, and ×1 where the base learning rate is 5e-4) and weight decay (×0.2, ×0.4, and ×1 where the base weight decay is 0.05). Note that the DeiT training is unstable when we use a larger learning rate or weight decay than the base values. Finally, we collect 22 `ResNet-50` models with different training regimes: 1 model with standard training by PyTorch (Paszke et al., 2019); 4 models trained by GluonCV (Guo et al., 2020)[2]; a semi-supervised model and semi-weakly supervised model on billion-scale unlabeled images by Yalniz et al. (2019)[3]; 5 models trained by different augmentation methods (Cutout (DeVries & Taylor, 2017), Mixup (Zhang et al., 2018), manifold Mixup (Verma et al., 2019), CutMix (Yun et al., 2019), and feature CutMix[4]; 10 optimized ResNet models by (Wightman et al., 2021)[5]. We also collect 7 `ViT-S` models with different training regimes, including the original ViT training setup (Dosovitskiy et al., 2021)[6], a stronger

---

[2]gluon_resnet50_v1b, gluon_resnet50_v1c, gluon_resnet50_v1d, and gluon_resnet50_v1s from `timm` library.
[3]ssl_resnet50 and swsl_resnet50 from `timm` library.
[4]We use the official weights provided by https://github.com/clovaai/CutMix-PyTorch.
[5]We use the official weights provided by https://github.com/rwightman/pytorch-image-models/releases/tag/v0.1-rsb-weights

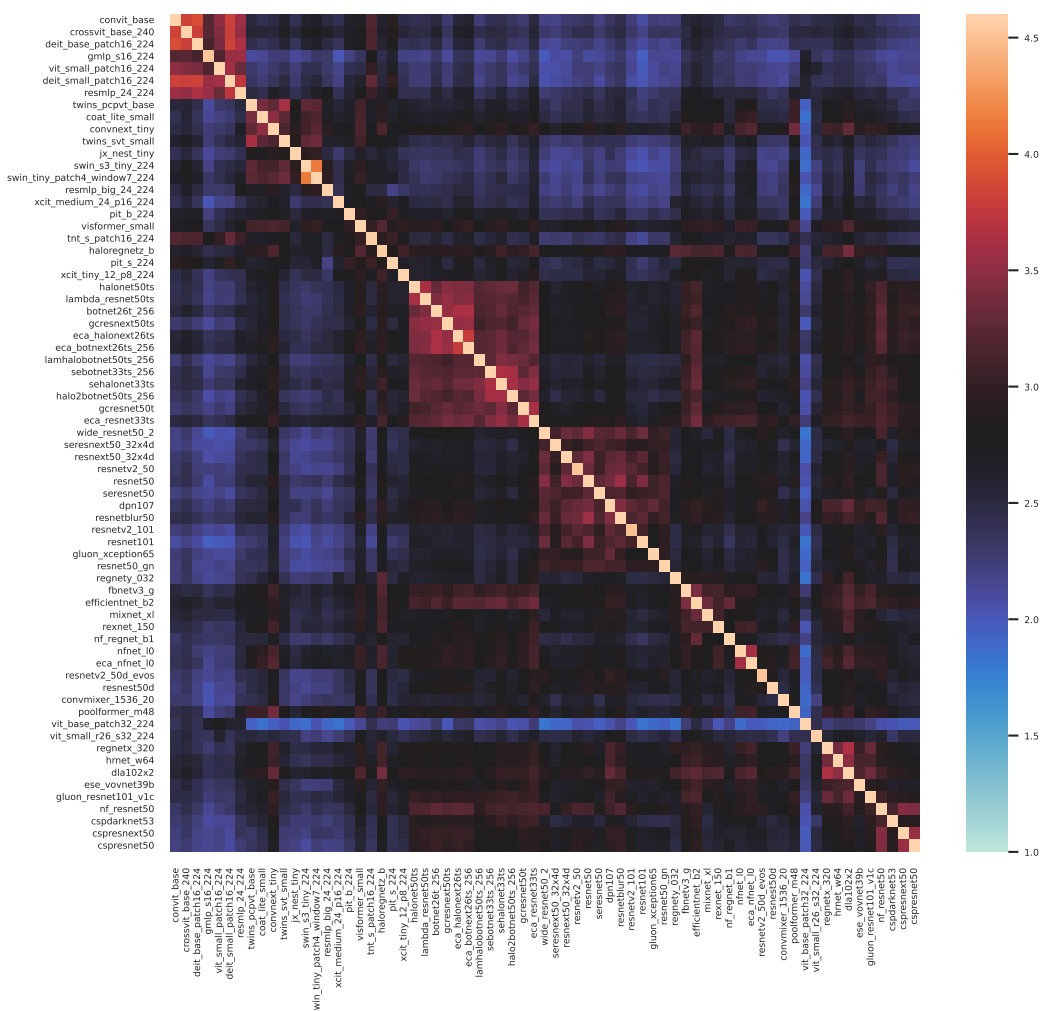

Figure D.2: **Pairwise similarity among 69 models.** Rows and columns are sorted by the clustering index in Tab. 2. $(n, n)$ component of pairwise similarity is close to 4.6 ($\log 100$) because the attack success rate is almost 100% when attacked model and a model used to generate are the same.

data augmentation setup in the Deit paper (Touvron et al., 2021)-3[6], the training setup with distillation (Touvron et al., 2021)-3[6], an improved DeiT training setup (Touvron et al., 2022b)-3[6], and self-supervised training fashions by MoCo v3 (Chen et al., 2021c)[7], MAE (He et al., 2022)[8] and BYOL (Grill et al., 2020)[9]. We do not use adversarially-trained networks because the adversarial training usually drops the standard accuracy significantly (Tsipras et al., 2019).

---

[6] deit_small_patch16_224, vit_small_patch16_224, deit_small_distilled_patch16_224, and deit3_small_patch16_224 from timm library.

[7] We train the `ViT-S` model by following https://github.com/facebookresearch/moco-v3

[8] We train the `ViT-S` model by following https://github.com/facebookresearch/mae

[9] We train the `ViT-S` model by following https://github.com/lucidrains/byol-pytorch

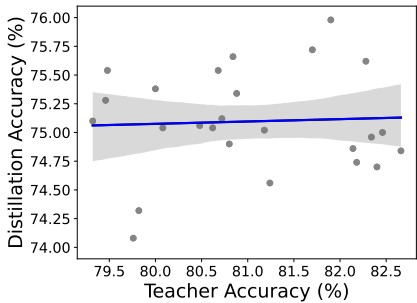

Figure F.1: **Teacher network accuracy and distillation performance.** The trend line and its 90% confidence interval are shown in the blue line and gray band. There is no significant correlation between teacher accuracy and distillation performance.

## F   KNOWLEDGE DISTILLATION

### F.1   TRAINING SETTING DETAILS FOR SEC. 4.2

We train `ViT-Ti` student models with 25 different teacher models using hard distillation strategy. We follow the distillation training setting of DeiT official repo[10], only changing the teacher model. Note that we resize the input images to the input size the teacher model requires if the input sizes of student and teacher models differ. If a teacher model needs a different input resolution, such as $240 \times 240$ and $256 \times 256$, we resize the input image for distilling it. Because `DeiT-Ti` has low classification accuracy compared to teacher models, the similarity score is calculated between `DeiT-S` and 25 models. The 25 teacher models are as follows: `BoTNet-26`, `CoaT-Lite Small`, `ConViT-B`, `ConvNeXt-B`, `CrossViT-B`, `CSPDarkNet-53`, `CSPResNeXt-50`, `DLA-X-102`, `DPN-107`, `EfficientNet-B2`, `FBNetV3-G`, `GC-ResNet-50`, `gMLP-S`, `HaloRegNetZ`, `MixNet-XL`, `NFNet-L0`, `PiT-S`, `RegNetY-032`, `ResMLP-24`, `ResNet-50`, `SEHaloNet33`, `Swin-T`, `TNT-S`, `VisFormer-S`, and `XCiT-T12`.

### F.2   TEACHER ACCURACY AND DISTILLATION PERFORMANCE

The similarity between teacher and student networks may not be the only factor contributing to distillation. For example, a stronger teacher can lead to better distillation performance (Yun et al., 2021). In Fig. F.1, we observe that if the range of the teacher accuracy is not significantly large enough (*e.g.*, between 79.5 and 82.5), then the correlation between teacher network accuracy and distillation performance is not significant; 0.049 Pearson correlation coefficient with 0.82 p-value. In this case, we can conclude that the teacher and student networks similarity contributes more to the distillation performance than the teacher performance.

## G   LIMTATIONS AND DISCUSSIONS

**Robustness to the choice of adversarial attack methods.**   Our similarity score is based on attack transferability using the representative gradient-based attack method, PGD attack (Madry et al. (2018)). To explore the effect of the choice of attack methods, we compute similarity scores on two different gradient-based attack methods; Autoattack (Croce & Hein (2020)), which is the state-of-the-art method attack method based on the ensemble of four diverse attacks, and Patchfool attack (Fu et al. (2022)), which is specially designed for vision transformers. We sample 8 models among 69 models for testing the effect of Autoattack on the similarity score: `ViT-S`, `CoaT-Lite Small`, `ResNet-101`, `LamHaloBotNet50`, `ReXNet-150`, `NFNet-L0`, `Swin-T` and `Twins-pcpvt`. Figure G.1a shows the high correlation between similarity scores using PGD and Autoattack; it shows a correlation coefficient of 0.98 with a p-value of

---

[10]https://github.com/facebookresearch/deit.

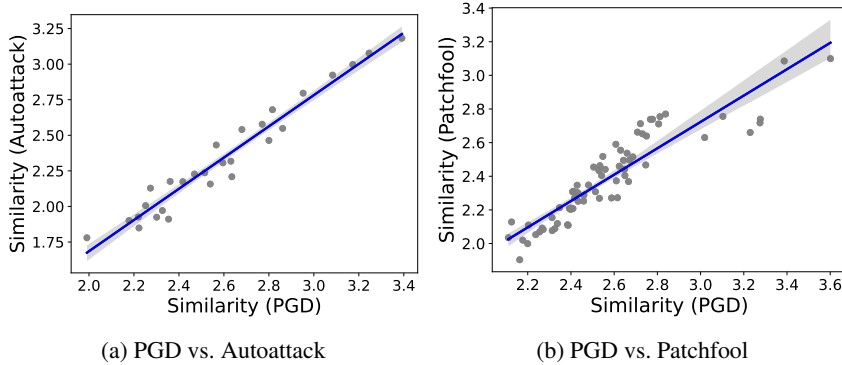

(a) PGD vs. Autoattack  (b) PGD vs. Patchfool

Figure G.1: **Effect of different adversarial attack methods to similarity score.** The trend line and its 90% confidence interval are shown in the blue line and gray band. We show the relationship between our similarity scores using PGD and other attacks, (a) Autoattack and (b) Patchfool attack.

$1.43 \times 10^{-18}$. For testing the Patchfool attack, we only generate adversarial perturbations on `ViT-S` and get attack transferability to all other models (68 models) because Patchfool targets transformer-based models. In the results, Patchfool also shows a high correlation compared to the PGD attack (Pearson correlation coefficient 0.91 with p-value $3.62 \times 10^{-27}$). The results show that our similarity score is robust to the choice of adversarial attack methods if the attack is strong enough. Note that PatchFool needs a heavy modification on the model code to extract attention layer outputs manually. On the other hand, PGD and AutoAttack are model-agnostic and need no modification on the model code. Therefore, if the PatchFool attack and the PGD attack show almost similar similarity rankings, we would like to recommend using PGD because it is easier to use for any architecture.

**Similarity scores for adversarially trained networks.** Because our method is based on an adversarial attack, it could be affected by an adversarial attack-aware learning method. We first note that our analysis is invariant to the adversarial training; as shown by Tsipras et al. (2019), adversarial robustness and accuracy are in a trade-off, and there is no ImageNet model with accuracy aligned with our target models yet. Also, our model similarity is based on the difference in the change of the model predictions. As shown by Tsipras et al. (2019) and Ilyas et al. (2019), adversarial training will lead to a different decision boundary. Therefore, we think that although our score can be affected by adversarial training, our score is still valid for adversarially trained models because an adversarially trained model is actually dissimilar to other models in terms of the decision boundary and representation learning.

**An efficient approximation for a new model.** We can use our toolbox for designing a new model; we can measure the similarity between a novel network and existing $N$ architectures; a novel network can be assigned to clusters (Tab. 2) to understand how it works. However, it requires generating adversarial samples for all $N+1$ models (*e.g.*, 70 in our case), which is computationally inefficient. Instead, we propose the approximation of equation 1 by omitting to compute the accuracy of the novel network on the adversarial samples of the existing networks. It will break the symmetricity of our score, but we found that the approximated score and the original score have high similarity (0.82 Pearson correlation coefficient with almost 0 p-value) as shown in Figure G.2.

As an example, we tested `Gluon-ResNeXt-50` (Guo et al., 2020), which has the same architecture with `ResNeXt-50` (Xie et al., 2017) in 69 models, and the distilled version of `DeiT-S` model (Touvron et al., 2021). Because the difference is not significant as we observed in Tab. D.1 and Fig. 4, we expect that `Gluon-ResNeXt-50` is assigned to the same cluster with `ResNeXt-50`, and distilled `DeiT-S` is assigned to the same cluster with `DeiT-S`. We observe that, as we expected, each network is assigned to the desired cluster. Therefore, we suggest using our efficient approximation for analyzing a novel network with our analysis toolbox.

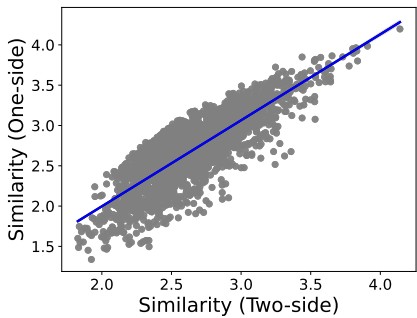

Figure G.2: **Similarity scores using two-side attack transferability vs. one-side attack transferability.** The trend line and its 90% confidence interval are shown in the blue line and gray band. We show the relationship between similarity scores using two-side and one-side attack transferability.

**Other possible applications.** We can use our similarity to develop a novel neural architecture or training technique by measuring how the proposed method can learn different representations compared to existing methods. Our similarity also can be used for analyzing the hyper-parameter sensitivity. Instead of reporting the performance changes by different hyper-parameters, one can report the average similarity of the trained models with different hyper-parameters. If the average similarity is significantly low, then we can argue that the method is sensitive to hyper-parameter selection. We expect this scenario is especially useful for the task, sensitive to the hyper-parameter selection (Choe et al., 2022; Gulrajani & Lopez-Paz, 2021). Finally, our similarity can be used for the model selection criteria for selecting diverse models, *e.g.*, constructing a new dataset with diverse pre-trained models for annotation candidates (Kuznetsova et al., 2020; Chun et al., 2022).

## H  ADVERSARIAL ATTACK TRANSFERABILITY AND DECISION BOUNDARY

Adversarial attack transferability is highly related to the decision boundaries of models. More specifically, measuring adversarial attack transferability is a good approximation of measuring the difference between model decision boundaries Karimi & Tang (2020). Comparing decision boundaries of two models will provide a high-level understanding of how two models behave differently for the changes of input. Unfortunately, an exact decision boundary is not achievable and an expensive and inexact sampling approximation is required (Somepalli et al., 2022). For example, Somepalli et al. (2022) uses a sampling-based approximation with 2,500 samples. More specifically, Somepalli et al. (2022) generates 2,500 perturbed samples (expanded by a coordinate defined by a randomly chosen triplet of inputs) for a given input and measures the ratio of "agreement" of two models (*e.g.*, if both two models predict 1,250 perturbed samples to the same label, then the similarity of decision boundaries for the given input is 1250/2500 = 0.5). However, it needs an expensive sampling approximation procedure and the approximation bound of the sampling method is not guaranteed. On the other hand, we can analyze the difference between two decision boundaries by generating an adversarial sample for each decision boundary. If two models have similar decision boundaries, then the adversarial samples will be transferred (*i.e.*,, will have high adversarial attack transferability); if two models have dissimilar decision boundaries, then the adversarial samples will not be transferred. Compared to the sampling-based approximation, adversarial attack transferability is more efficient (we used 50 iterations for the attack, while sampling needs 2,500 samples) and less stochastic than a sampling-based approximation. Also, we employed an adversarial attack based on gradient descent where the convergence of gradient descent is theoretically guaranteed, while sampling approximation has no guarantee on its approximation error. Therefore, adversarial attack transferability can work as a good proxy measure of the difference of model decision boundaries.

