# OpenReview forum: "Similarity of Neural Architectures Based on Input Gradient Transferability"
_ICLR.cc/2023/Conference — Submitted to ICLR 2023_

### Official Review · Reviewer_kjGm · 2022-10-21

**Confidence:** 4
**Correctness:** 3
**Technical Novelty And Significance:** 2
**Empirical Novelty And Significance:** 3
**Recommendation:** 5

**Clarity, Quality, Novelty And Reproducibility:**

The paper is well-written and easy to follow. The analysis is sound and reasonable. I did not identify obvious obstacles in reproducing the results, however, providing the code can improve the reproducibility of this work.

**Strength And Weaknesses:**

__Strength__

__[S1] Novel model similarity measure.__ Using attack transferability of adversarial examples to measure model similarity is relatively novel. As a data-oriented approach, the similarity measure can be easily applied to different model architectures.

__[S2] The analysis reveals some interesting findings.__ I appreciate the comprehensive analysis of the model diversity. The analysis identifies some factors that lead to more diverse models measured by attack transferability. The findings may be of interest to some data partitioners in the community.

__[S3]__ Overall, the paper is easy to follow.

__Weakness__

__[W1] Some findings are mostly expected or have been studied before.__
- Intuitively, an adversarial example generated for a specific model is more effective on the same base architecture. For example, using the adversarial examples generated from ResNet to attack VGG is more difficult than attacking other ResNet models. So, it is somewhat expected that the proposed method considers the differences in model architecture as the most diverse change.
- The observation that using more diverse models in ensemble learning leads to better performance has been explored in many previous works.
- As suggested by the paper, the observation that using a similar model architecture for student and teacher models results in better performance is studied in previous KD works.

__[W2] Limitation of using attack transferability as similarity measures.__ As using the adversarial attack transferability naturally gives a higher similarity score for the same model architecture, is it possible that the proposed similarity measures give an incorrect similarity score in terms of the predictions:
- Given two models with the same base architecture, the models make very different predictions; however, the adversarial examples transfer well across the two models.
- Given two models with different base architectures, the models make very similar predictions; however, the adversarial examples do not transfer well across the two models.

__[W3] Does not discuss and compare with other diversity measures.__ Although the proposed similarity measures can improve ensemble learning, the paper does not compare the approach with previously suggested model diversity metrics in ensemble learning literature, e.g. [1]. It is difficult to judge whether the proposed method is a better measurement in selecting the models in ensemble learning.

__Other comments__

__[O1]__ If the set of candidate models contains defended models that are trained to be robust to adversarial samples, does the proposed similarity measure also work?

__[O2]__ The paper uses PGD as the adversary. Is the method sensitive to the choice of attacks?

__Ref__

[1] Kuncheva, Ludmila & Whitaker, Chris. (2003). Measures of Diversity in Classifier Ensembles and Their Relationship with the Ensemble Accuracy. Machine Learning. 51. 181-207. 10.1023/A:1022859003006.


**Summary Of The Paper:**

The paper proposes using the attack transferability of adversarial examples to measure neural architecture similarity. The authors analyze the design choices leading to better model diversity measured by the proposed metric. The paper also demonstrates the application of the proposed similarity measure in ensemble learning and knowledge distillation.

**Summary Of The Review:**

This paper is well-written and proposes a simple and practical similarity measure for neural architecture. However, this paper lacks the discussion and comparison with the model diversity metric proposed in previous works. The paper should show and explain why the attack transferability metric is a better measurement of model similarity.

---

> ### Author Response · Authors · 2022-11-16
> **Response to Reviewer kjGm (1/4)**
>
> We appreciate the positive and constructive comments by Reviewer kjGm, e.g., our paper is easy to follow, proposed similarity score using attack transferability is novel, and our analysis is comprehensive and reveals some interesting findings.
>
> We first explain the motivation and insights for why adversarial attack transferability can serve as a similarity function, and we will address all questions and concerns raised by Reviewer kjGm.
>
> ### Insights and motivations of using adversarial attack transferability for defining the similarity function
>
> First, we would like to explain why adversarial attack transferability (e.g., `accA→B + accB→A`) can act as a similarity function. Ideally, we would like to define a similarity function between two models with the following three properties: (1) $x = \arg\min_y d(x, y)$ (2) $d(x,y) = d(y,x)$ (3) $d(x, y) > d(x, x)$ if $x \neq y$. The adversarial attack transferability (e.g., `accA→B + accB→A`) generally satisfies these properties. If we assume the adversary is optimal, then accA→A will be zero and it will be the minimum (because accuracy is non-negative). `accA→B + accB→A` is symmetric. It will satisfy $d(x, y) \geq d(x, x)$ if $x \neq y$ where it is a weaker condition than (3). In other words, adversarial attack transferability can serve as a similarity score function from a functional point of view.
>
> Second, we use adversarial attack transferability as the proxy measure of the difference between input gradients of two networks. Input gradient is a widely-used framework to understand model behavior, e.g., how a model will change predictions by local pixel changes. Our motivation is that if two models are more similar, then the input gradients are more similar. However, because input gradient is very noisy, directly measuring the difference between input gradients is also very noisy. Instead, we generate adversarial samples using input gradients (e.g., using PGD, AutoAttack). Because adversarial samples are updated by input gradients, adversarial samples will be similar if input gradients of two networks are similar. Therefore, we expect that high adversarial attack transferability denotes two networks have similar input gradients.
>
> Finally, adversarial attack transferability is highly related to the decision boundaries of models. More specifically, measuring adversarial attack transferability is a good approximation of measuring the difference between model decision boundaries [1]. Comparing decision boundaries of two models will provide a high-level understanding of how two models behave differently for the changes of input. Unfortunately, an exact decision boundary is not achievable and an expensive and inexact sampling approximation is required [2]. For example, [2] uses a sampling-based approximation with 2,500 samples. More specifically, [2] generates 2,500 perturbed samples (expanded by a coordinate defined by a randomly chosen triplet of inputs) for a given input and measures the ratio of “agreement” of two models (e.g., if both two models predict 1,250 perturbed samples to the same label, then the similarity of decision boundaries for the given input is 1250/2500 = 0.5). However, it needs an expensive sampling approximation procedure and the approximation bound of the sampling method is not guaranteed. On the other hand, we can analyze the difference between two decision boundaries by generating an adversarial sample for each decision boundary. If two models have similar decision boundaries, then the adversarial samples will be transferred (i.e., will have high adversarial attack transferability); if two models have dissimilar decision boundaries, then the adversarial samples will not be transferred. Compared to the sampling-based approximation, adversarial attack transferability is more efficient (we used 50 iterations for the attack, while sampling needs 2,500 samples) and less stochastic than a sampling-based approximation. Also, we employed an adversarial attack based on gradient descent where the convergence of gradient descent is theoretically guaranteed, while sampling approximation has no guarantee on its approximation error. Therefore, adversarial attack transferability can work as a good proxy measure of the difference of model decision boundaries.
>
> - [1] Karimi, Hamid, and Jiliang Tang. "Decision boundary of deep neural networks: Challenges and opportunities." Proceedings of the 13th International Conference on Web Search and Data Mining (2020).
> - [2] Gowthami Somepalli, et al., “Can neural nets learn the same model twice? investigating reproducibility and double descent from the decision boundary perspective”, CVPR (2022).
>
> Our initial manuscript slightly mentioned the first and the second insights. We will revise our manuscript to emphasize and clarify the motivation and insights for why adversarial attack transferability can serve as the similarity function.

---

> > ### Author Response · Authors · 2022-11-16
> > **Response to Reviewer kjGm (2/4)**
> >
> > ### [W1] On some findings in this paper
> >
> > Our main contribution is focused on the similarity metric itself and analysis based on the proposed similarity metric. We also do not argue that our finding is the first observation that shows more diversity leads better ensemble performance or that shows more similar teacher-student architectures lead to a better KD performance. We agree that these observations are already discussed before, but in fact, these observations strengthen our proposed similarity score; the similarity measured by our score function is aligned with the observations from previous literature (By the way, it will be very grateful if the reviewer can provide more related works. We would like to cite them if we missed any).
> >
> > Furthermore, we would like to emphasize that our finding in KD is novel; previous literature partially shows that a more similar teacher-student pair leads to better/worse KD performances (Section 4.2), however, our finding shows that this statement depends on the architecture type of teacher and student; if both teacher and student are based on Transformer. then a more similar pair leads a better KD performance; on the other hand, if the teacher is not a Transformer, then the Transformer student will show a better performance with a dissimilar teacher.
> >
> > Finally, although the reviewer’s first comment (“intuitively, an adversarial example generated for a specific model is more effective on the same base architecture”) could be intuitive, we would like to emphasize that our work is the first work to show the intuition actually works in practice with a fairly large (69) menu of models with different architectural components (as in the Reviewer ynKk’s comment). Especially, our work is the first work that analyzes the actual contribution of components on the diversity based on the statistical analysis. Although a statement could be intuitive and sometimes looks trivial, it is worth to show the statement actually holds in a scientific way. We show that the existing scattered findings (as the reviewer listed) can be explained through the lens of the model diversity in terms of the adversarial attack transferability.
> >
> > We will revise our paper to clarify our contribution.
> >
> > ### [W2] Extreme cases of measuring similarity using attack transferability
> >
> > Thanks for pointing out two extreme cases. It will be very helpful for the readers to show the cases can be handled by our similarity score. Before describing each case, we first clarify what we expect for each case. We define our similarity score based on the concept of input gradient and decision boundary (as our first comment). We also assume that “similar predictions” denotes that the prediction vectors (e.g., Softmax outputs) are similar. Note that we focus on the change of predictions. Different predictions will show different amounts of change of predictions. However, similar predictions are not guaranteed to show similar amounts of the change of predictions. Intuitively speaking, even if two functions have the same output value at point $x$, their gradient values at point $x$ can be significantly different (e.g., f(x) = x and g(x) = x^2 will have the same function x=1: `f(1) = g(1) = 1`, but different gradient values at x=1: `f’(1) = 1` and `g’(1)=2`.
> >
> > **Case 1. Same architecture, but very different predictions**
> >
> > In short, our similarity score shows low similarities in this case. In Section 3.2, we show that different training regimes lead to low similarity scores. We collect 22 ResNet-50 models including the standard PyTorch model, 4 GlounCV-trained models, a semi-supervised model, and semi-weakly supervised model on billion-scale unlabeled images by Yalniz et al. (2019), 5 models trained by different augmentation methods (Cutout, Mixup, manifold Mixup, CutMix, and feature CutMix), 10 optimized ResNets by Wightman et al., 2021. We also collect 7 ViT-S models including the original ViT, DeiT, DeiT-distilled, DeiT-v3, MoCo v3-trained ViT, MAE-trained ViT and BYOL-trained ViT. Not all of these models work significantly different from each other, but we can expect that many of them will work differently.
> >
> > As shown in Table 3, “Training regime” shows lower similarity scores (3.27/3.44 for ResNet and ViT) than the models with different initialization (4.23/4.21 for ResNet and ViT) and hyper-parameters (3.27/3.44 for ResNet and ViT). The average similarity between all 69 models is 2.73. This shows that even though two models are based on the same architecture, if their decision boundaries differ significantly, then our similarity score will be low.

---

> > > ### Author Response · Authors · 2022-11-16
> > > **Response to Reviewer kjGm (3/4)**
> > >
> > > **Case 2. Different architectures, but very similar predictions**
> > >
> > > As we mentioned before, because we focus on the change of predictions, simply showing the similar predictions will not guarantee low or high similarity. Especially, considering that recent DNN models sometimes show very similar predictions [3], we argue that just capturing the difference of prediction values is not enough to measure the similarity between models.
> > >
> > > [3] Robert Geirhos, et al., “Trivial or impossible--dichotomous data difficulty masks model differences (on ImageNet and beyond)”, ICLR (2022)
> > >
> > > ### [W3] Discussion about other diversity measures
> > >
> > > We compare previous other network comparison works to our method in Section 2, including Kornblith et al 2019, Raghu et al 2021, Geirhos et al 2018, Geirhos et al 2020, Scimeca et al 2022, Dinh et al 2017, Li et al 2018, Park & Kim 2022, Somepalli et al 2022. In the paper, we mentioned that these works are mainly focused on understanding model differences by visualization or other ad-hoc analysis, not quantitative score. Our contribution is to propose a quantitative measure of the network similarity and to show three analyses (component-level contribution, ensemble vs. similarity, and KD vs. similarity). We thank Reviewer kjGm to introduce an interesting related work (Kuncheva et al [R1]). We discuss how our method and Kuncheva et al is different here, and will add related discussion in the revised paper.
> > >
> > > [R1] Kuncheva, Ludmila & Whitaker, Chris. (2003). Measures of Diversity in Classifier Ensembles and Their Relationship with the Ensemble Accuracy. Machine Learning. 51. 181-207. 10.1023/A:1022859003006.
> > >
> > > We first note that comparing different similarity measures is extremely difficult because there is no ground-truth of similarities. For example, assume there exists similarity function A and B. Can we say A is better than B if ensemble performances and A has a better correlation than B? There could be many similarity scores focusing on different properties, and the ensemble performance could be one of the possible properties of diversity. We emphase that Kuncheva et al also denoted that it is difficult to judge that which measure is the best measure (from Section 5 and Conclusion of [R1]):
> > >
> > > > We have to be cautious when designing diversity measures based on oracle outputs. The measures should not be a replacement for the estimate of the accuracy of the team but should stem from the intuitive concept of diversity.
> > >
> > > > It is difficult to judge which measure best expresses the concept of diversity, which leaves question 4, as stated in the introduction, open for further debate.
> > >
> > > > A choice of measure to recommend can be based on ease of interpretation. (...) The use of diversity measures for enhancing the design of classifier ensembles (question 5) is still an open question. Given the lack of a definitive connection between the measures and the improvement of the accuracy, it is unclear whether measures of diversity will be of any practical value at this stage
> > >
> > > Kuncheva et al [R1] also recommended $Q_{av}$ among 10 prediction-based statistics because it is easy to calculate and easy to interpret, and it has 0 value for independence.  It shows that directly comparing two (or more) similarity scores is extremely difficult. For this reason, we did not compare our method to other similarity scores.
> > >
> > > Also, the diversity measures in Kuncheva et al [R1] are based on model predictions. However, as we mentioned in [W2], we focus on the change of predictions, not the prediction itself. Because recent deep models are highly complex, the change of predictions (or sensitivity) is more practical than focusing on the prediction itself. For example, a deep model easily changes its prediction with a neglected small noise (e.g., adversarial attack). It shows that just focusing on prediction values could be misleading. Therefore, we argue that a similarity score based on the change of predictions is better than a similarity score based on prediction solely.
> > >
> > > We did not argue that our similarity score is the best similarity score. Furthermore, we did not argue that our similarity score is designed for ensemble learning (or knowledge distillation). We aim to propose a new similarity score between two models with a quantitative value, and our analysis shows an example of practical applications of our similarity score.

---

> > > > ### Author Response · Authors · 2022-11-16
> > > > **Response to Reviewer kjGm (4/4)**
> > > >
> > > > Finally, our goal is focused on proposing a new similarity score based on the input gradient view-point (and decision boundary view-point as mentioned in [Insights and motivations of using adversarial attack transferability for defining the similarity function]) and showing practical applications of our similarity score (e.g., component contribution analysis, similarity vs. ensemble, similarity vs. KD). We did not argue that our similarity score is the best, or our method will improve ensemble learning. Our experiments show that our similarity score works well as we expected and as previous work supported. As far as we know, our work is the first work to show various applications of a similarity function on a fairly large (69) menu of models with different architectural components (As comments by Reviewer ynKk and Reviewer UduE). We believe that our contribution mainly lies in the extensive analysis with 69 state-of-the-art deep models, not proposing the best similarity score.
> > > >
> > > > We will revise our paper accordingly.
> > > >
> > > > ### [O1] If the set of candidate models contains defended models that are trained to be robust to adversarial samples, does the proposed similarity measure also work?
> > > >
> > > > As we mentioned in [W2] Case 1, we believe that two models with the same architecture are dissimilar if their decision boundaries significantly differ. Also as we mentioned in the first comment, adversarial attacks and decision boundaries are highly related. In other words, the adversarially trained model will have a significantly different decision boundary from the other models. Therefore, we think that it will show a low similarity with the other models.
> > > >
> > > > In practice, we omit adversarially trained models in our study because (1) there exists a trade-off between adversarial robustness and standard accuracy [4], (2) we limit the models with similar standard training methods and with similar accuracies, and (3) we focus on practical models, while an adversarially trained network is not yet practical (due to low accuracy [4] and high computational resources).
> > > >
> > > > [4] Tsipras, Dimitris, et al. "Robustness may be at odds with accuracy." ICLR (2019).
> > > >
> > > > ### [O2] The paper uses PGD as the adversary. Is the method sensitive to the choice of attacks?
> > > >
> > > > We compare our method with two different attack methods: AutoAttack [5] and PatchFool [6]. AutoAttack is the state-of-the-art attack method based on an ensemble of four diverse attacks. PatchFool is an attack method specifically designed for Transformers. As shown in Figure G.1 (in the revised version), the attack transferabilities by AutoAttack and PatchFool are highly correlated to the attack transferability by PGD (Pearson correlation coefficients are 0.91 and 0.98, respectively.). It means that our similarity score is robust to the choice of adversarial attack methods if the attack is strong enough. Note that PatchFool needs a heavy modification on the model code to manually extract attention layer outputs, on the other hand, PGD and AutoAttack are model-agnostic attacks and need no modification on the model code. Therefore, if the PatchFool attack and the PGD attack show almost similar similarity rankings, we would like to recommend using PGD because it is easier to use for any architecture.
> > > >
> > > > - [5] Croce, Francesco, and Matthias Hein. "Reliable evaluation of adversarial robustness with an ensemble of diverse parameter-free attacks." ICML (2020).
> > > > - [6] Fu, Yonggan, et al. "Patch-Fool: Are Vision Transformers Always Robust Against Adversarial Perturbations?." ICLR (2022).
> > > >
> > > > ### Providing the code can improve the reproducibility of this work.
> > > >
> > > > We will release our codebase publicly for reproducibility in the future.

---

> > > > > ### Author Response · Authors · 2022-11-25
> > > > > **Thanks for your comment again**
> > > > >
> > > > > Dear Reviewer kjGm,
> > > > >
> > > > > Thanks for your constructive reviews and efforts for the ICLR community. We wonder whether our revision and response have addressed your concerns.
> > > > >
> > > > > We revised our paper to clarify our motivation and insights for using adversarial transferability. We also clarified that previous findings actually support our observations and similarity score. We also added a discussion about other diversity measures, e.g., Kuncheva et al. Finally, our additional experiments showed that our similarity score is robust to the choice of adversarial attack.
> > > > >
> > > > > The detailed changes in the revision are highlighted in red texts. We are happy to discuss any further questions if you have any.

---

> > > > > > ### Comment · Reviewer_kjGm · 2022-11-28
> > > > > > **Post-Rebuttal Comments**
> > > > > >
> > > > > > I would like to sincerely thank the authors for the detailed response. The author explained why adversarial transferability could be used as a similarity function and clarified some limitations. I appreciate these responses, but I tend to keep the current score due to the following concerns:
> > > > > >
> > > > > > - The authors explained the differences between using adversarial transferability and model predictions to measure model similarities. However, the advantages of the proposed method are not very clear. For example, the authors argued that: "Because recent deep models are highly complex, the change of predictions (or sensitivity) is more practical than focusing on the prediction itself." It would be better if the claim was supported with more evidence.
> > > > > >
> > > > > > - I agree with the authors that it is difficult to compare different similarity measures as there are no ground-truth similarities. However, it is still useful to include some diversity baselines and compare them with the proposed method in different applications, so we will know which approach is better under which scenarios.
> > > > > >
> > > > > > - As the paper does not show the advantages of the proposed method over existing diversity measures, it is hard to justify the use of the method in practice, especially since it requires additional computation to generate the adversarial examples.

---

> > > > > > > ### Author Response · Authors · 2022-12-09
> > > > > > > **Additional Response to Reviewer kjGm (1/2)**
> > > > > > >
> > > > > > > Thanks for your comments. We will address all raised concerns by Reviewer kjGm.
> > > > > > >
> > > > > > > We emphasize the conceptual advantages of adversarial transferability compared to prediction only methods for two reasons. First, as in our previous rebuttal comment, we focus on the difference in the change of the predictions, not the difference in the predictions. As the same prediction scores can have different changes in predictions (as in our previous comment), the change in the predictions will have more information than the predictions. It is because predictions only express classification results, rather than focusing on decision making or model behavior.
> > > > > > >
> > > > > > > As in our previous comment, we use adversarial attack transferability as the proxy measure of the difference between input gradients of two networks, where measuring adversarial attack transferability is a good approximation of measuring the difference between model decision boundaries [1]. Our method has more benefits than an expensive and inexact sampling approximation [2] as shown in our previous comment (Insights and motivations of using adversarial attack transferability for defining the similarity function).
> > > > > > >
> > > > > > > - [1] Karimi, Hamid, and Jiliang Tang. "Decision boundary of deep neural networks: Challenges and opportunities." Proceedings of the 13th International Conference on Web Search and Data Mining (2020).
> > > > > > > - [2] Gowthami Somepalli, et al., “Can neural nets learn the same model twice? investigating reproducibility and double descent from the decision boundary perspective”, CVPR (2022).
> > > > > > >
> > > > > > > As the second advantage of adversarial transferability, we note that the classification performances are getting almost perfect (e.g., 100%) as deep models are getting more complex [3]. In other words, predictions are getting similar to each other especially for nearly perfect models, and in this case prediction-based methods will be not informative than decision boundary-based methods. Similar to the extreme cases of Reviewer kjGm’s first comment, we could suggest another extreme case: Given two models with different basic architectures, the models have similar predictions because their performances are almost 100%. In that case, prediction-based measurements are not meaningful anymore while our measurements can show the differences. Furthermore, [3] already showed that recent DNN models have very similar predictions on the ImageNet dataset. We expect this comment will resolve the reviewer’s first comment.
> > > > > > >
> > > > > > > [3] Robert Geirhos, et al., “Trivial or impossible--dichotomous data difficulty masks model differences (on ImageNet and beyond)”, ICLR (2022)
> > > > > > >
> > > > > > > We also conducted additional experimental results for comparing our method to other similarity measures. While our measurement shows a reasonable similarity between models, prediction-based measurement (we use the first statistical method described in [R1]) shows less intuitive results. For example, the similarity score based on model predictions between DeiT-S and ViT-S is 49-th place among similarity scores between DeiT-S and other 68 models, while it is 5th place in the case of our similarity score. Note that ViT and DeiT are the same architecture, where DeiT is trained with more stable and data-efficient training strategies than ViT. Therefore, judging that ViT and DeiT are dissimilar is not an intuitive result.
> > > > > > >
> > > > > > > Furthermore, we observe that the prediction-based similarity score [R1] shows vulnerability to changes in a model scale. We evaluate whether models having the same basic architecture (e.g., DeiT-B and DeiT-S) show a similar tendency of similarity score to other models or not. We gain similarity scores between DeiT-B (ResNet50) or DeiT-S (ResNet101) and other models except for the two models (i.e., 67 models) and calculate the Pearson correlation coefficient between them. In the result, the prediction-based similarity score shows low correlation (Pearson correlation coefficients are 0.83 and 0.85 for DeiT and ResNet, respectively), while our similarity score shows very high correlation (Pearson correlation coefficients are 0.97 and 0.98 for DeiT and ResNet, respectively). We expect our new experimental result will resolve the reviewer’s second and third concerns.
> > > > > > >
> > > > > > > [R1] Kuncheva, Ludmila & Whitaker, Chris. (2003). Measures of Diversity in Classifier Ensembles and Their Relationship with the Ensemble Accuracy. Machine Learning. 51. 181-207. 10.1023/A:1022859003006.

---

> > > > > > > > ### Author Response · Authors · 2022-12-09
> > > > > > > > **Additional Response to Reviewer kjGm (2/2)**
> > > > > > > >
> > > > > > > > Regarding the reviewer’s third concern (additional computation), we hope that our previous comment in “Insights and motivations of using adversarial attack transferability for defining the similarity function” can resolve the concern even for the decision boundary-based methods:
> > > > > > > >
> > > > > > > > > Unfortunately, an exact decision boundary is not achievable and an expensive and inexact sampling approximation is required [2]. For example, [2] uses a sampling-based approximation with 2,500 samples. More specifically, [2] generates 2,500 perturbed samples (expanded by a coordinate defined by a randomly chosen triplet of inputs) for a given input and measures the ratio of “agreement” of two models (e.g., if both two models predict 1,250 perturbed samples to the same label, then the similarity of decision boundaries for the given input is 1250/2500 = 0.5). However, it needs an expensive sampling approximation procedure and the approximation bound of the sampling method is not guaranteed. On the other hand, we can analyze the difference between two decision boundaries by generating an adversarial sample for each decision boundary. If two models have similar decision boundaries, then the adversarial samples will be transferred (i.e., will have high adversarial attack transferability); if two models have dissimilar decision boundaries, then the adversarial samples will not be transferred. Compared to the sampling-based approximation, adversarial attack transferability is more efficient (we used 50 iterations for the attack, while sampling needs 2,500 samples) and less stochastic than a sampling-based approximation. Also, we employed an adversarial attack based on gradient descent where the convergence of gradient descent is theoretically guaranteed, while sampling approximation has no guarantee on its approximation error. Therefore, adversarial attack transferability can work as a good proxy measure of the difference in model decision boundaries.
> > > > > > > >
> > > > > > > > Finally, we would like to emphasize that our work is focused on the analysis of differences in model architectures. The reasons mentioned above show the limitations of usage prediction-based similarity measures to the analysis of model architectures. For the same reason, we could not observe meaningful results on distillation. We will revise our paper to contain this context before the camera-ready due.

---

> > > > > > > > > ### Author Response · Authors · 2022-12-13
> > > > > > > > > **Any following question from Reviewer kjGm?**
> > > > > > > > >
> > > > > > > > > Dear Reviewer kjGm,
> > > > > > > > >
> > > > > > > > > We wonder whether our revision and response have addressed your concerns. We revised our paper to clarify our motivation and insights for using adversarial transferability. Our additional response also contains more detailed insights. We also clarified that previous findings actually support our observations and similarity score. Our additional response shows that other similarity scores have disadvantages compared to our approach both conceptually and empirically. We also added a discussion about other diversity measures, e.g., Kuncheva et al. Finally, our additional experiments showed that our similarity score is robust to the choice of adversarial attack.
> > > > > > > > >
> > > > > > > > > Please don't hesitate to bother us if you have any further questions.

---

### Official Review · Reviewer_yYKW · 2022-10-23

**Confidence:** 4
**Correctness:** 1
**Technical Novelty And Significance:** 2
**Empirical Novelty And Significance:** 3
**Recommendation:** 3

**Clarity, Quality, Novelty And Reproducibility:**

The presentation is clear, logically coherent and related works are comprehensively covered. It starts with motivation and problem statement, followed by literature review and the proposed similarity score. Then it continues with experimental evaluation and outline of practical scenarios. The perspective of using transferability of input gradients via adversarial examples are novel and the decomposition of DNN into 13 components provide useful guideline for future studies. The reproducibility appears to be good as well. The paper would be much strengthened if authors propose the similarity score for CNN and Transformer separately based on different mechanisms of generating adversarial examples.

**Strength And Weaknesses:**

The problem of measuring DNN similarity sees a broad range of applications in model diversity, such as choosing diverse models for ensemble, and similar models for knowledge distillation. The categorization of DNN into 13 different components provides a useful guideline for future studies.

A major weakness to the reviewer lies in the rationale of defining the similarity score, i.e., transferability of adversarial examples generated by PGD. The underlying assumption states if the adversarial examples are transferrable between a pair of DNNs, then the similarity is high. It is more reasonable if the similarity is measured between a pair of CNNs but not so between a pair of Transformers nor between a pair of CNN and Transformer. This is because the adversarial examples generated by PGD on CNN may not be transferrable to Transformer, evident by several recent publications on transferability of adversarial examples (e.g., Mahmood et al, ICCV-2021, Fu et al, ICLR-2022). In fact, the algorithms to generate adversarial examples for Transformer can never be the same as those for CNNs. This is because Transformer operates on image patches to extract global features via self-attention whereas CNN operates on pixels to extract local features via convolution feature maps. As such, the reviewer believes the similarity score is limited to CNN on image classification tasks.


**Summary Of The Paper:**

This paper proposed a new similarity function to quantify the similarity of the DNNs in image classification tasks based on transferability of adversarial examples. For a pair of similar DNNs, authors investigate which types of DNN components contribute most to the similarity and their implications to practical scenarios. Using a collection of pre-trained image classifiers on ImageNet, authors conclude that the base architecture component (i.e., convolution-based CNN vs. self-attention-based Transformer) plays a most important role in measuring similarity and cluster DNNs according to the 13 categories of DNN components.

**Summary Of The Review:**

This paper tackles an important problem of measuring DNN similarity in terms of image classification task. The similarity score is well motivated by the input transferability and practical application in model diversity, either increase diversity in generating ensembles or decrease diversity in selecting teacher and student in knowledge distillation, are also promising. However, the major technical concern is the similarity score may be more suitable for measure CNN similarities. For Transformer similarities, a different mechanism to generate adversarial are required.

---

> ### Author Response · Authors · 2022-11-16
> **Response to Reviewer yYKW (1/2)**
>
> We appreciate the constructive comments by Reviewer yYKW. We also thank the reviewer for positive feedback, e.g., clear and logically coherent presentation, comprehensively covered related works, and usefulness of our component analysis. As far as we understood, the reviewer’s main concern is that “our similarity between Transformers and CNNs will show a low similarity score **because** CNNs and Transformers show a low transferability”, and “the origin of the low transferability between Transformers and CNNs is the attack mechanism for Transformers should be different from CNNs”. We will address all concerns raised by Reviewer yYKW.
>
> ### The similarity score may be more suitable for measure CNN similarities
>
> The objective function of an adversarial attack can be written as:
> $$ \arg\max_\{x, \| x - x_0 \| \leq \varepsilon \} \left\[\mathcal L( f(x), y)\right\],$$
> for a given network parameter $f$, a ground-truth label $y$, an initial input $x_0$, and small $\varepsilon$. Here, the PGD attack seeks an input $x$ that maximizes the loss function (e.g., cross entropy) using the projected gradient descent optimization algorithm. Note that the PatchFool attack (Fu et al. 2022 [1]) also uses the same objective function, but it uses a patch-wise optimization algorithm specialized to Transformers. In Fu et al. 2022, a strong attack method such as PGD or AutoAttack still can achieve almost zero accuracy for Transformer (e.g., PGD shows 2.28% attacked accuracy for DeiT-B), while PatchFool is stronger than PGD (0% for DeiT-B). As our purpose is not to achieve a perfect attack but to use adversarial attack for measuring a similarity between models,we believe PGD is strong enough for our purpose.
>
> [1] Fu, Yonggan, et al. "Patch-Fool: Are Vision Transformers Always Robust Against Adversarial Perturbations?." ICLR (2022).
>
> In addition, we show that the adversarial attack transferabilities by PGD and PatchFool [1] are highly correlated. More specifically, we measure the similarities between DeiT and the other models using PGD and PatchFool, respectively. In the revised paper Figure G.1, we can observe that attack transferabilities by PatchFool and PGD are highly correlated (Pearson correlation coefficient 0.91 with p-value 3.62e-27). We also show that AutoAttack also shows a high correlation to PGD (Pearson correlation coefficient 0.98 with p-value 1.43e-18). It means that our similarity score is robust to the choice of adversarial attack methods if the attack is strong enough. Note that PatchFool needs a heavy modification on the model code to manually extract attention layer outputs, on the other hand, PGD and AutoAttack are model-agnostic and need no modification on the model code. Therefore, if the PatchFool attack and the PGD attack show almost similar similarity rankings, we would like to recommend using PGD because it is easier to use for any architecture.
>
> Also, we do not think the observation by Mahmood et al 2021 [2] violates our findings. For example, Mahmood et al. stated that
>
> >  In general, the phenomenon of low transferability mostly occurs between model genusus, but not within model genusus. That is to say, adversarial examples generated by one BiT model will likely transfer to a different BiT model, but not to a ViT model or ResNet.
>
> Our main observation in Sections 3.1 and 3.2 also shows the same phenomenon. As far as we understood, Mahmood et al. did not argue that Transformers and CNNs have low attack transferability because of the attack method. Note that the proposed ensemble attack by [2] is for attacking an ensemble system constructed by models with low attack transferability (e.g., ResNet + ViT), not designed for a single model.
>
> [2] Mahmood, Kaleel, Rigel Mahmood, and Marten Van Dijk. "On the robustness of vision transformers to adversarial examples." ICCV (2021).
>
> As Reviewer yYKW mentioned, we agree to the origin of the low attack transferability is the architectural differences (such as stem-layer design or the existence of self-attention layer as). This could be supported by our previous experimental result: ConvNeXt is a CNN but it shows high similarity with Twins, CoaT, and Swin (Table 2 cluster No 2). If CNN and Transformers always show a low attack transferability due to the attack mechanism, ConvNeXt should show a low similarity with Twins, CoaT, and Swin.

---

> > ### Author Response · Authors · 2022-11-16
> > **Response to Reviewer yYKW (2/2)**
> >
> > ### Summary
> >
> > As far as we understood, the reviewer’s comment means that “our similarity between Transformers and CNNs will show a low similarity score **because** CNNs and Transformers show a low transferability”, and “the origin of the low transferability between Transformers and CNNs is the attack mechanism for Transformers should be different from CNNs”. However, we think in a reverse way. First, we think the attack mechanism is not a problem. We prove this by showing a high correlation between the PatchFool (a Transformer-specific attack method) transferability and the PGD transferability. Second, as Reviewer yYKW mentioned, we also think the origin of the low attack transferability is the architectural differences (such as stem-layer design or the existence of self-attention layer). This could be supported by our previous experimental result: ConvNeXt is a CNN but it shows high similarity with Twins, CoaT, and Swin (Table 2 cluster No 2). If CNN and Transformers always show a low attack transferability due to the attack mechanism, ConvNeXt should show a low similarity with Twins, CoaT, and Swin.

---

> > > ### Author Response · Authors · 2022-11-25
> > > **Thanks for your comment again**
> > >
> > > Dear Reviewer yYKW,
> > >
> > > Thanks for your valuable service to the ICLR community. We wonder whether our revision and response have addressed your concerns.
> > >
> > > Especially, we have shown that our method is invariant to the choice of the architecture (e.g., CNN or Transformer) by showing that the similarity measured by the PatchFool attack and the similarity measured by the PGD attack have a high correlation. Our response also has addressed the reviewer's main concern.
> > >
> > > The detailed changes in the revision are highlighted in red texts. We are happy to discuss any further questions if you have any.

---

> > > > ### Author Response · Authors · 2022-12-13
> > > > **Any following question from Reviewer yYKW?**
> > > >
> > > > Dear Reviewer yYKW,
> > > >
> > > > We wonder whether our revision and response have addressed your concerns. Especially, we have shown that our method is invariant to the choice of the architecture (e.g., CNN or Transformer) by showing that the similarity measured by the PatchFool attack and the similarity measured by the PGD attack have a high correlation. Our response also has addressed the reviewer's main concern.
> > > >
> > > > Please don't hesitate to bother us if you have any further questions.

---

### Official Review · Reviewer_Bk5e · 2022-10-23

**Confidence:** 4
**Correctness:** 1
**Technical Novelty And Significance:** 1
**Empirical Novelty And Significance:** 1
**Recommendation:** 1

**Clarity, Quality, Novelty And Reproducibility:**

Clarity

This paper is often confusing. The writing and organization need to be significantly improved, e.g. "then accA→B will not be dropped significantly"

Quality

The paper is technically weak, and it is hard to tell what contributions it makes.

Novelty

The novelty of this paper is not clear. For example, "we found that more diversity leads to better ensemble performance" has been recognized in ensemble learning for a long time.

Reproducibility

Good.

**Strength And Weaknesses:**

Strength

+ Similarity of DNN architecture is an important research topic.


Weaknesses

- What does "If A and B are similar and assume an optimal adversary, then accA→B will be almost zero" mean?

- There are numerous syntax errors. e.g., please revise "We expect our analysis tool helps a high-level understanding of differences between various neural architectures as
well as practical guidance when using multiple architectures."

- What is "destructive success"?

- Please elaborate "distinct properties".

- How is similarity score defined?

- What is input gradient?

**Summary Of The Paper:**

This paper present a DNN model similarity measure using input gradient transferability. The basic hypothesis is that if two neural networks are similar, adversarial attack transferability will be high. Additionally two topics are investigated: (1) Which network component contributes to the model diversity? (2) impact of model diversity in practice.

**Summary Of The Review:**

This paper is lack of technical contribution and novelty.

---

> ### Author Response · Authors · 2022-11-16
> **Response to Reviewer Bk5e (1/2)**
>
> We appreciate the comments by Reviewer Bk5e.
>
> Before addressing the questions raised by Reviewer Bk5e, we re-emphasize our technical contribution and novelty.
>
> As our first contribution, we propose a similarity score function between two neural networks. Note that unlike previous network similarity analysis tools, our method can provide the exact number that satisfies the essential properties of similarity function (Please check the answer for the first question for details).
>
> Our second and third contributions are focused on analysis based on the proposed score function. Our analysis is not only for ensemble analysis but we also show component-wise contribution analysis and KD analysis.
>
> We do not argue that our finding is the first observation that shows more diversity leads better ensemble performance. In fact, our observation strengthens our proposed similarity score; the similarity measured by our score function is aligned with the observations from previous literature. We mentioned it in the revised paper. It will be our pleasure if Reviewer Bk5e provides references if we missed any related work.
>
> Furthermore, we would like to emphasize that our finding in KD is novel; previous literature partially shows that a more similar teacher-student pair leads to better/worse KD performances (Section 4.2), however, our finding shows that this statement depends on the architecture type of teacher and student; if both teacher and student are based on Transformer, then a more similar pair leads a better KD performance; on the other hand, if the teacher is not a Transformer, then the Transformer student will show a better performance with a dissimilar teacher.
>
> We also emphasize that our work is the first work to show the intuition actually works in practice with a fairly large (69) menu of models with different architectural components (as in the Reviewer ynKk’s comment)
>
> Finally, we would like to emphasize that all other Reviewers (Reviewers ynKk, UduE, yYKW and kjGm) agreed with that our paper is well-written and easy to follow. We would like to expect that Reviewer Bk5e will revise the reviewer’s opinion about the clarity and quality by our answers.
>
> Now, we address all questions raised by Reviewer Bk5e.
>
> ### What does "If A and B are similar and assume an optimal adversary, then accA→B will be almost zero" mean?
>
> **accA→B** means the accuracy of model B for the input attacked by adversarial perturbation generated for model A. For example, acc{AlexNet→ResNet} means the accuracy of ResNet when the inputs are the adversarially attacked samples containing perturbations optimized for AlexNet.
>
> Ideally, we would like to define a similarity function between two models with the following three properties: (1) $x = \arg\min_y d(x, y)$ (2) $d(x,y) = d(y,x)$ (3) $d(x, y) > d(x, x)$ if $x \neq y$. The sentence raised by Reviewer Bk5e is for illustrating a high level understanding of our similarity score; the sentence recaps that accA→B satisfies the first and the third properties. As we define our similarity score by accA→B + accB→A, our similarity score also satisfies the second property as well.
>
> **Optimal adversary** is an adversarial attack method that makes accX→X always zero for any model X. Note that our method does not need an optimal adversary, but we would like to show that if we have an optimal adversary (a very strong adversarial attack method), then accA→B becomes zero, as many distance functions have nearly zero values for very similar two inputs.
>
> We modify the section to clarify our statement.

---

> > ### Author Response · Authors · 2022-11-16
> > **Response to Reviewer Bk5e (2/2)**
> >
> > ### How is similarity score defined?
> >
> > The full definition and description of our similarity score are given in Section 2. Especially, the definition of our similarity score is as follows (Equation 1 in our submission):
> >
> > $$s(A,B) = \log \left\[ 100 \times \frac{1}{2|X_{AB}|} \sum_{x \in X_{AB}} ( \mathbb {I} ({A(x_B)} \neq y) + \mathbb {I} ({B(x_A)} \neq y) ) \right\]$$
> >
> > Here, $X_{AB}$ denotes the set of validation samples where both A and B predict the correct labels, and $\mathbb {I} ({A(x_B)} \neq y)$ denotes the number of wrong predictions by A when the adversarial samples generated for B are given. We take the logarithm function because the score can have a small value.
> >
> > For example, let’s assume two models A and B, and the test set X where $|X|$ = 50,000. We first seek $X_{AB}$ where both networks predict the correct labels. Let’s assume $|X_{AB}|$ = 40,000. Next, we generate adversarially attacked samples for A and B on $X_{AB}$. Then, we compute the error of network A (or B) for the adversarially attacked samples for B (or A), and take the average. Let’s assume $\mathbb {I} ({A(x_B)} \neq y)$ = 12,000 and $\mathbb {I} ({B(x_A)} \neq y)$ = 14,500. Finally, we take the logarithm function to get the score. In our example, it will be log [100 * 1/40,000 * 1/2 * (12,000 + 14,500) ] = log (33.125) $\approx$ 3.50
> >
> > ### What is input gradient?
> > Input gradient is the gradient regarding to inputs, namely, $\frac{\partial \mathcal L}{\partial x}$ where $\mathcal L$ is a loss function and $x$ is an input. It can be computed by the chain-rule (or backpropagation) as follows = $\frac{\partial \mathcal L}{\partial x} = \frac{\partial \mathcal L}{\partial f_N} \frac{\partial f_N}{\partial f_{N-1}} \ldots \frac{\partial f_2}{\partial f_1} \frac{\partial f_1}{\partial f_x}$. In terms of the PyTorch code, it is easily obtainable as follows:
> >
> > ```
> > loss = cross_entropy(model(x), label)
> > model.backward()
> > input_gradient = x.grad()
> > ```
> > Note that the dimensionality of input gradient is the same as the dimensionality of the original input. For example, if an input is a 3x224x224 image, then the input gradient is also 3x224x224. Input gradient is widely-used for understanding models because it denotes how sensitive a model is to the input changes. Our work aims to measure the difference of input gradients by utilizing adversarial attaks. Note that not all adversarial attacks utilize input gradient, but the state-of-the-art attacks, such as PGD and AutoAttack are based on input gradient: iteratively update an input using the input gradient `x + x.grad()` to seek an input making a worse loss value.
> >
> > ### Please elaborate "distinct properties".
> > We aim to understand diverse properties by different neural networks. There could be many properties, for example, there could be a “texture-biased” network that only focusing on local texture of the given input, while there also could be a “shape-biased” networks focusing on global shape information [1]. If we can quantify the similarity between given two models, then we can measure how different design choices (different component choices, training strategies, ...) make the different properties. We will revise the sentence for a clarification.
> >
> > [1] Geirhos, Robert, et al. "ImageNet-trained CNNs are biased towards texture; increasing shape bias improves accuracy and robustness." ICLR (2019).
> >
> > ### What is "destructive success"?
> > As Reviewer ynKk also pointed out the same sentence, and we also agree with the reviews that the sentence is confusing. We fix the word to “great success” to make the sentence clearer.
> >
> > ### Syntax errors and clarity
> > Thanks for pointing out. We re-check and revise our manuscript focusing on grammar errors and improving clarification.

---

> > > ### Author Response · Authors · 2022-11-25
> > > **Thanks for your comment again**
> > >
> > > Dear Reviewer Bk5e,
> > >
> > > We wonder whether our revision and response have addressed your concerns and questions. The detailed changes in the revision are highlighted in red texts. We are happy to discuss any further questions if you have any.

---

> > > > ### Author Response · Authors · 2022-12-13
> > > > **Any following question from Reviewer Bk5e?**
> > > >
> > > > Dear Reviewer Bk5e,
> > > >
> > > > We wonder whether our revision and response have addressed your concerns. Please don't hesitate to bother us if you have any further questions.

---

### Official Review · Reviewer_UduE · 2022-10-24

**Confidence:** 4
**Correctness:** 3
**Technical Novelty And Significance:** 3
**Empirical Novelty And Significance:** 3
**Recommendation:** 6

**Clarity, Quality, Novelty And Reproducibility:**

This paper is well written and easy to understand. The proposed method is of good novelty.

**Strength And Weaknesses:**

Strength:
1. This paper proposes a quantitative similarity score between different neural architectures based on the adversarial attack transferability.
2. The proposed smiliarity helps to understand the component-level architecture design, and leads to better understanding of the relationship between model similarity of model ensemble performance and model distillation performance.
3. Several interesting observations are obtained based on the proposed similarity function.
4. Extensive experiments and analysis are conducted, to lead to better understanding.

Weaknesses:
1. Could the authors provide more motivation and insights for why  adversarial attack transferability can serve as the similarity function?
2. In Figure 6 (c), most teacher models are clustered together with quite similar slimilarity. More teachers with large variety of similarity should be provided to better justify the conclusion that using a more dissimilar teacher leads to better distillation performance.

**Summary Of The Paper:**

This paper proposes a quantitative similarity score between different neural architectures based on the adversarial attack transferability. This smiliarity helps to understand the component-level architecture design, and leads to better understanding of the relationship between model similarity of model ensemble performance and model distillation performance. Several interesting observations are obtained based on the proposed similarity function.

**Summary Of The Review:**

This paper is interesting and leads to several observations that may be beneficial to architecture design, model ensemble and knowledge distillation. More analysis and insights for the proposed similarity function should be provided. Several experiments related to Figure 6 are encouraged to be conducted.

---

> ### Author Response · Authors · 2022-11-16
> **Response to Reviewer UduE (1/2)**
>
> We deeply appreciate the valuable and positive comments by Reviewer UduE, e.g., our paper is well written, our similarity score is helpful to understand component-level architecture design and the relationship between model similarity and ensemble/KD, and our observations based on extensive experiments and analysis are interesting. We will address all concerns raised by the reviewer and revise the paper accordingly.
>
> ### Could the authors provide more motivation and insights for why adversarial attack transferability can serve as the similarity function?
>
> First, we would like to explain why adversarial attack transferability (e.g., `accA→B + accB→A`) can act as a similarity function. Ideally, we would like to define a similarity function between two models with the following three properties: (1) $x = \arg\min_y d(x, y)$ (2) $d(x,y) = d(y,x)$ (3) $d(x, y) > d(x, x)$ if $x \neq y$. The adversarial attack transferability (e.g., `accA→B + accB→A`) generally satisfies these properties. If we assume the adversary is optimal, then accA→A will be zero and it will be the minimum (because accuracy is noative). `accA→B + accB→A` is symmetric. It will satisfy $d(x, y) \geq d(x, x)$ if $x \neq y$ where it is a weaker condition than (3). In other words, adversarial attack transferability can serve as a similarity score function from a functional point of view.
>
> Second, we use adversarial attack transferability as the proxy measure of the difference between input gradients of two networks. Input gradient is a widely-used framework to understand model behavior, e.g., how a model will change predictions by local pixel changes. Our motivation is that if two models are more similar, then the input gradients are more similar. However, because input gradient is very noisy, directly measuring the difference between input gradients is also very noisy. Instead, we generate adversarial samples using input gradients (e.g., using PGD, AutoAttack). Because adversarial samples are updated by input gradients, adversarial samples will be similar if input gradients of two networks are similar. Therefore, we expect that high adversarial attack transferability denotes two networks have similar input gradients.
>
> Finally, adversarial attack transferability is highly related to the decision boundaries of models. More specifically, measuring adversarial attack transferability is a good approximation of measuring the difference between model decision boundaries [1]. Comparing decision boundaries of two models will provide a high-level understanding of how two models behave differently for the changes of input. Unfortunately, an exact decision boundary is not achievable and an expensive and inexact sampling approximation is required [2]. For example, [2] uses a sampling-based approximation with 2,500 samples. More specifically, [2] generates 2,500 perturbed samples (expanded by a coordinate defined by a randomly chosen triplet of inputs) for a given input and measures the ratio of “agreement” of two models (e.g., if both models predict 1,250 perturbed samples to the same label, then the similarity of decision boundaries for the given input is 1250/2500 = 0.5). However, it needs an expensive sampling approximation procedure and the approximation bound of the sampling method is not guaranteed. On the other hand, we can analyze the difference between two decision boundaries by generating an adversarial sample for each decision boundary. If two models have similar decision boundaries, then the adversarial samples will be transferred (i.e., will have high adversarial attack transferability); if two models have dissimilar decision boundaries, then the adversarial samples will not be transferred. Compared to the sampling-based approximation, adversarial attack transferability is more efficient (we used 50 iterations for the attack, while sampling needs 2,500 samples) and less stochastic than a sampling-based approximation. Also, we employed an adversarial attack based on gradient descent where the convergence of gradient descent is theoretically guaranteed, while sampling approximation has no guarantee on its approximation error. Therefore, adversarial attack transferability can work as a good proxy measure of the difference in model decision boundaries.
>
> - [1] Karimi, Hamid, and Jiliang Tang. "Decision boundary of deep neural networks: Challenges and opportunities." Proceedings of the 13th International Conference on Web Search and Data Mining. 2020.
> - [2] Gowthami Somepalli, et al., “Can neural nets learn the same model twice? investigating reproducibility and double descent from the decision boundary perspective.” CVPR (2022)
>
> Our initial manuscript slightly mentioned the first and the second insights. We will revise our manuscript to emphasize and clarify the motivation and insights for why adversarial attack transferability can serve as the similarity function.

---

> > ### Author Response · Authors · 2022-11-16
> > **Response to Reviewer UduE (2/2)**
> >
> > ### Distribution of similarity scores of teacher models in Figure 6 (c)
> >
> > Thanks for the comment. We first remark that Figure 6 (c) already includes all non-ViT teachers that have a similarity score larger than 2.5. We sample 25 models among 67 models (except for ViT and DeiT) according to the distribution of the similarity scores to prevent the sampling bias. Note that a low similarity score denotes that a teacher network is dissimilar to the ViT student. Because Transformers and other networks (e.g., CNNs) differ significantly, most of CNN teachers (Figure 6c) have a lower similarity score to the ViT student than Transformer teacher similarities. We carefully re-check the models in timm and other public repository, but we couldn’t find more proper teacher networks that statisfy our model selection criterion (whose top-1 accuracy is between 79% and 83%). We will add a related discussion to the revised paper.

---

> > > ### Author Response · Authors · 2022-11-25
> > > **Thanks for your comment again**
> > >
> > > Dear Reviewer UduE,
> > >
> > > Thanks for your valuable efforts to ICLR. We wonder whether our revision and response have addressed your concerns.
> > >
> > > We especially revised our paper focusing on motivation and insights of our method (as our response). We also clarify Figure 6 (c).
> > >
> > > The detailed changes in the revision are highlighted in red texts. We are happy to discuss any further questions if you have any.

---

> > > > ### Author Response · Authors · 2022-12-13
> > > > **Any following question from Reviewer UduE?**
> > > >
> > > > Dear Reviewer UduE,
> > > >
> > > > We wonder whether our revision and response have addressed your concerns. (e.g., we especially revised our paper focusing on motivation and insights of our method as our response, and we also clarified Figure 6 (c)). Please don't hesitate to bother us if you have any further questions.

---

### Official Review · Reviewer_ynKk · 2022-10-25

**Confidence:** 4
**Correctness:** 4
**Technical Novelty And Significance:** 3
**Empirical Novelty And Significance:** 3
**Recommendation:** 8

**Clarity, Quality, Novelty And Reproducibility:**

- Clarity: Paper is well written and easy to understand
- Quality: Paper includes a fairly detailed study of the proposed similarity metric, with a few interesting insights (e.g. choice of stem has a large impact on similarity).
- Novelty: Reasonable. Paper repurposes adversarial perturbations to define a similarity and presents novel conclusions on choice of teacher.
- Reproducibility: The proposed metric is fairly simple. If the images used to compute similarity are shared, then result should be reproducible.


**Strength And Weaknesses:**

Strengths
- Well written and easy to understand paper
- A detailed analysis of the similarity measure w.r.t. a fairly large (69) menu of models with different architectural components.
- Interesting insights into effect of component choices on similarity and ensembling performance.

Weaknesses
- Proposed method is computationally expensive and the approximation proposed in 5 is rather ad-hoc and not well evaluated. This limits the applicability of the method.

Minor nitpick
- What do authors mean by "destructive success of DNNs"? The phrase seems to refer to negative effects of DNN success. Perhaps, rephrase is that is not intended, otherwise clarify.


**Summary Of The Paper:**

The paper proposes a method to compute pairwise similarities between two architectures. This is done by evaluating the change in prediction of first model caused by the adversarial prediction w.r.t. second model and viceversa, with the intuition being that if two models are similar then their adversarial perturbations would have similar effect. Paper then uses the proposed similarity measure to analyze existing model architecture choices and effect on ensembling performance.

**Summary Of The Review:**

Overall, I feel that the technical contribution is fairly limited. However, the paper presents an interesting an fairly detailed study that provides interesting insights and would be beneficial to a large audience. I vote for accept.

---

> ### Author Response · Authors · 2022-11-16
> **Response to Reviewer ynKk**
>
> We are very grateful for the positive and constructive reviews by Reviewer ynKk, e.g., our paper is well written and easy to understand, our analysis is fairly large and detailed, and our findings on component analysis and model ensemble are interesting. We will address all concerns raised by the reviewer and revise the paper accordingly.
>
> ### About the computational cost of our method
>
> First, we would like to emphasize that our method focuses on measuring a quantitative similarity between two models on *a given validation dataset* while the existing network similarity literature focuses on qualitative visualizations for *a given image* (not the whole validation dataset) [1].
>
> [1] Kornblith, Simon, et al. "Similarity of neural network representations revisited", ICML (2019)
>
> As far as we know, [2] is the only work that proposes a quantitative similarity function of two networks (based on decision boundary). [2] also needs high computational resources because an exact decision boundary is not achievable so it is approximated by a sampling procedure. For example, [2] uses a sampling-based approximation with 2,500 samples. On the other hand, we set the iteration of the PGD attack to 50 (50 forward operations and 50 backward operations), which is $\approx$25 times more efficient than 2,500 samplings (2,500 forward operations). Note that our method is based on gradient descent whose convergence is theoretically guaranteed, while [2] is based on a sampling-based approximation without any upper bound of the number of samples; we would like to emphasize that our approach is more efficient in terms of the iterations.
>
> [2] Gowthami Somepalli, et al., “Can neural nets learn the same model twice? investigating reproducibility and double descent from the decision boundary perspective”, CVPR (2022)
>
> ### Evaluation of the approximation proposed in Section 5
>
> Thanks for the constructive comment. The approximation assumes a situation when there already exists a pre-computed N x N similarity matrix, and a new model comes. Because our method needs adversarial attacks on both networks, it needs (N + 1) adversarial attacks. Instead, the approximation only computes (1) adversarial attack by omitting to compute the accuracy of the new model on the adversarially attacked samples to the existing networks. The example of the initial manuscript shows that even if we approximate similarities, the clustering works similarly as we expect.
>
> However, we agree with Reviewer ynKk that the approximation was not evaluated well; so, to address the reviewer’s concern, we have verified the approximation actually works. Here, we show that the approximation (attacking only the new network and evaluating the existing networks using the attacked samples) does not harm the similarity rankings and is statistically meaningful. We have compared two rankings -- (1) a ranking sorted by our proposed symmetric score (2) a ranking sorted by their approximated score -- for 69 models. In average, we observed that (1) and (2) are statistically similar; it shows 0.82 Pearson correlation score. We will add a related discussion to the revised paper soon.
>
> ### Generated images
>
> Thanks for the comment! We also considered publicly releasing the generated images. However, because we should not use efficient lossy compression methods (such as JPEG compression) to keep the adversarial effect, we have to save an image as a 3 x 224 x 224 matrix, instead of the efficient JPEG. Unfortunately, it makes the size of the whole generated images too huge to save (about 207.73 GB = (4byte)*(3*224*224)*5000*69). Instead, we will release our codebase publicly for reproducibility.
>
> ### Minor comment
>
> We fix “destructive success of DNNs” to “great success of DNNs”. We agree that the initial sentence is confusing and unclear.

---

> > ### Author Response · Authors · 2022-11-25
> > **Thanks for your comment again**
> >
> > Dear Reviewer ynKk,
> >
> > Thanks for your valuable service to the ICLR community. We wonder whether our revision and response have addressed your concerns. (e.g., we added a detailed evaluation for our approximation in Section 5).
> >
> > The detailed changes in the revision are highlighted in red texts. We are happy to discuss any further questions if you have any.

---

> > > ### Author Response · Authors · 2022-12-13
> > > **Any following question from Reviewer ynKk?**
> > >
> > > Dear Reviewer ynKk,
> > >
> > > We wonder whether our revision and response have addressed your concerns. (e.g., we added a detailed evaluation for our approximation in Section 5). Please don't hesitate to bother us if you have any further questions.

---

> > > > ### Comment · Reviewer_ynKk · 2022-12-13
> > > > **Satisfied with the responses**
> > > >
> > > > I am satisfied with the responses to my computational complexity concerns and am glad to see further evaluations for 5. After going through other reviews and responses, I see concerns about novelty by other reviewers. I am surprised to see those concerns as my impression was that the proposed similarity measure is novel. Further, a fairly comprehensive study provides evidence that indeed the proposed similarity measure works as expected and also suggests novel conclusions about choice of teacher. I will maintain my accept rating.

---

### Author Response · Authors · 2022-11-16
**Revision Plan**

Dear reviewers and AC,

Thank you for your thoughtful comments and valuable services to the community.

We are encouraged they found that our analysis is insightful (Reviewers ynKk, UduE, kjGm) and useful (Reviewer yYKW). We are pleased they found our method is novel (Reviewers ynKk, kjGm) and its topic, model similarity, is important. They found our experiments are based on fairly many models (Reviewer ynKk), so very extensive (Reviewer UduE), and all of these are clearly written (Reviewers ynKk, UduE, yYKW, kjGm).

By reflecting on reviews, we would like to revise our paper for better understanding and discussion. Here, we share our detailed revision plan for our paper.

### Insights and motivations of using adversarial attack transferability for defining the similarity function

We will add a detailed discussion related to insights behind our similarity score (as the comment for Reviewer UduE and kjGm). It will be added to the end of Section 2 based on the comments for Reviewer UduE and kjGm.

### Previous findings

We will add previous findings (such as ensemble vs. diversity) in Section 4. We will emphasize that our findings are aligned to prior results (e.g., diversity helps ensemble [1]) for each corresponding analysis.

[1] Kuncheva, Ludmila & Whitaker, Chris. (2003). Measures of Diversity in Classifier Ensembles and Their Relationship with the Ensemble Accuracy. Machine Learning. 51. 181-207. 10.1023/A:1022859003006.

### Comparisons with similarity scores with other attack methods.

We will add comparison results of our similarity score and its variants. We will show that our method is robust to the choice of attack method, such as PGD (used in the paper), AutoAttack (the state-of-the-art attack method), and PatchFool (a Transformer-specific attack method) by showing the rankings measured by each score are statistically similar. We will also add the correlation figure between our method and the approximated version of our method in Section 5. We will add this discussion in Section 5 and Appendix due to the space limitation.

### Discussion of why it is difficult to directly compare different similarty scores

As the comment for Reviewer kjGm, it is extremely difficult to compare different similarity scores with diverse properties. We will add related discussion in Section 2 and Appendix.

### Introduction

We will also include the above contents in the introduction section as well.

- We will clarify motivations and insights of using attack transferability for measuring model similarity. We will add it as we explain in individual responses.
- We will clarify our contribution by focusing on the similarity score and analysis using it. We will add discussion about findings already discovered and newly updated.
- We will add discussion how our work is different from additional related work suggested by Reviewer kjGm. Our work focuses on input gradient and decision boundary while the related work used model predictions. Also, we will clarify our goal is not developing a new method for improving ensemble performance, but measuring model diversity and analysis in the aspects of three viewpoints (component-level contribution, ensemble vs. similarity, and KD vs. similarity).

### Minor revisions

- destructive success => great success
- We will clarify several explanations (e.g., explaining distinct properties more precisely)
- We will re-check and revise our manuscript focusing on grammar errors and improving clarification.

### Additional experiments

Here, we contain additional experiments not included in the initial submission version:

- As mentioned in our response for Reviewer ynKk, we will revise the paper to contain the evaluation of the effect of the approximation, which uses only one-side attack transferability to measure similarity. We explored a relationship between similarity scores using two-side attack transferability and one-side attack transferability (Figure G.2). It shows that a one-side attack does not harm the overall tendency. We will add the experimental details and discussion soon.
- As mentioned in our response for Reviewer yYKW and kjGm, we will revise the paper to contain the robustness of our method to the choice of attack methods. We explored Patchfool attack (Figure G.1 (a)) and Autoattack (Figure G.1 (b)) and showed that they are highly correlated to PGD. We will add the experimental details and discussion.

---

### Decision · Program_Chairs · 2023-01-20

**Decision:**

Reject

**Justification For Why Not Higher Score:**

While the paper presents an interesting direction for studying the similarity of neural networks through adversarial input transferability, the reviewers question the theoretical justification of the proposed idea. Additionally, comparing the method against prior baselines and related methods, studying the method under controlled settings (e.g., toy examples), or analyzing the effect of adversarial training on the proposed proxy would help readers better understand why adversarial transferability is a good similarity proxy. Given these, the paper does not appear to be ready for publication at ICLR in its current form.

**Justification For Why Not Lower Score:**

N/A

**Metareview: Summary, Strengths And Weaknesses:**

This paper presents a neural network similarity proxy based on adversarial robustness transferability. Under this framework, if two networks have similar accuracy against each other's adversarial inputs, they are considered similar. The proposed similarity proxy is studied for ensemble classification and knowledge distillation. The contribution of different neural network components to the similarity proxy is also discussed.

Pros:
- The paper is written clearly and very nicely
- The similarity of an exhaustive list of 69 networks is studied

Cons:
- Using adversarial attack transferability as a similarity proxy is not theoretically grounded in this submission and it is not analyzed in controlled settings (e.g. toy datasets, simple networks, or known similarities)
- The effect of adversarial training itself on the proposed proxy is not studied
- The introduced method has a high computational cost (minor issue though)
- Reviewers found some of the findings less surprising
- Reviewers suggest that comparing to related baselines could strengthen the message of this paper

**Summary Of Ac-Reviewer Meeting:**

N/A